# Exploring the Role of Knowledge, Innovation and Technology Management (KNIT) Capabilities that Influence Research and Development

**Zeeshan Asim *** and **Shahryar Sorooshian**

Department of Industrial Management, Universiti Malaysia Pahang Lebuhray Tun Razak, Gambang, Kuantan 26300, Malaysia; sorooshian@gmail.com
* Correspondence: zeeshanasimump@gmail.com; Tel.: +601-2346-2861

**Abstract:** A supporting management discipline is an under researched phenomenon. A majority of firms' operating capabilities relating to knowledge, innovation and technology management as a supporting management discipline. This paper reviews the literature in the research and development (R&D) domain to formulate qand propose a conceptual model which is influenced by capabilities relating to knowledge, innovation and technology management. We performed a systematic literature review in which a range of articles were searched related to R&D, from 1990 to 2018. Our review is presented in two parts. The first part presents a descriptive analysis using the Preferred Reporting Items for Systemic View and Mata Analysis (PRISMA)protocol and the second part develop proposition based on keywords analysis. The review revealed that there are nine capabilities that contribute to influence on R&D based on three dimensions known as knowledge, innovation and technology management which correspond to some of the key resources that used to drive these capabilities. Finally, this work proposes a conceptual model based on the outcome of Systematic Literature Review (SLR) interpretation. This research may support current trends on the literature. The list of references may be considered a potential source for future research in this area.

**Keywords:** knowledge management; innovation management; technology management (KNIT); knowledge management capabilities (KMC); innovation management capabilities (IMC); technology management capabilities (TMC); R&D

## 1. Introduction

Why most research and development (R&D) are firms in developed countries more successful in adopting new capabilities to confront market dynamism as compared to firms in developing countries? Although the significance of R&D has long been acknowledged among developing countries and is considered a central stream element for developing science and technology policies to counter economic and social challenges [1], an amount of evidence suggests that most R&D firms in developed countries at government level enhanced their existing capabilities to capture the market change [2].

The role of R&D with reduced spending, remains highly encouraged in order to sustain business objectives and to drive national innovation systems in uncertain conditions [3]. There is no reservation regarding the significance of R&D among developing countries as a potential instrument for confronting growing challenges due to fast technological development at a large industrial scale [4]. However, many developing countries still face technical barriers due to slowing progress in their R&D [5].

Organization of Economic Co-operation and Development (OECD) [6] shows that the majority of governments in developing countries support potential resources for R&D in public organizations but still the outcomes of a majority of R&D in public firms were fragmented and disarticulated,

making them ineffective [7]. More than 60% of public organizations among a majority of developing countries have been unable to classify the relationship in between capabilities related knowledge, innovation and technology (KNIT) management as significant contributors to R&D [1,8,9]. Such issues frequently appeared in the form of market failure and systematic failure as outcomes of R&D [1,9,10]. The prior research more concerning about to rectify market and systematic failure also highlights some methods to overcome these failures [1,9,11–14], but did not address aspects of capabilities failures that appeared during the capability learning process among various public organizations which fail to categorize the relationship between the capabilities related to knowledge, innovation and technology management [1,8,15,16]. Therefore, classification of capabilities related to supporting a management discipline is needed to propel R&D to significant outcomes [17].

In this research, we used PRISMA (preferred reporting items for systematic review) technique along with co-word analysis to visualize the existing literature to address the question, 'Is there any relationship between the capabilities related to (KNIT) management that share their boundaries with R&D' which allow policy makers to develop a sustainable national innovation system?' We will merge results from all existing literature and scan common applicable outcomes. From the Novelty prospective, while previous literature has reflected some of the roles of R&D in different ways to measure a firm's competitive advantages and have been used as performance indicator, there is little argument available regarding the inter connectivity between these three sets of supporting management capabilities with their effective influence on R&D in order to avoid the difficulties of classifying the capabilities during the learning process.

The outcome of this research represents the conceptual model that fills the gap by connecting some of theories that individually represent the dimensions of knowledge, innovation and technology management capabilities. Such an interpretation highlights the roles of knowledge, innovation and technology management in supporting management discipline that contribute to an impact on R&D with supportive evidence relating to the factors on which these capabilities rely. These factors help to stimulate theoretical understanding in the form of criteria and sub-criteria that assist decision makers in how effective selection decisions are made and under what conditions these capabilities influence overall R&D.

The outline of this research article is as follows: Section 2 presents the literature gap, while Section 3 presents the PRISMA (preferred reporting items for systematic review) technique, Section 4 presents the co-word analysis along with the visualization of previous studies within the same domain, Section 5 presents some discussion. Finally, Section 6 presents the conceptual model and conclusions.

## 2. Why Is this Review Important?

Developing countries are looking to contribute more on R&D at public organizations because knowledge, innovation and technology (KNIT) management capabilities are vital for national competitiveness [18]. At a minimum level, these developing countries require aggressive capabilities for R&D in order to align with global industrial trends according to their local conditions [6].

However, major contributions to economic strength among these developing countries depend on R&D in the public sector which has so far confronted various capability failures due to infrastructural weakness in supporting disciplines [1]. A recent study by the World Management Survey (WMS) has legitimatized a quantum leap in the comparative study of the contribution of KNIT management capabilities as a supporting management discipline and their implications for R&D and industrial productivity [19]. WMS illustrates various market failures due to deficiencies in distinguishing capabilities related to KNIT management as supporting management disciplines [19]. Redressing these capability deficiencies emerges as a stressful and engaging process for many developing economics for sustaining their R&D [1]. Evidence suggests that, despite the substantial R&D myopia that occurs among various developing countries due to weak national innovation policy [19], ultimately the focus of various researchers is diverted to emphasizing the significance of a broader set of capabilities related to a supporting management discipline at the technological frontier [19]. The globalization of

economics has highlighted the significance of entrepreneurial action for creating more wealth [20,21]. Such global economic transformation allows firms to update their value-added capabilities which may stimulate a new composition of organizational strength [21]. It may significantly influence R&D at a governmental and entrepreneurial level in various developing countries [6,21]. As a result of such influence, entrepreneurs in multiple sectors face immense challenges in upgrading their value-added capability based on knowledge, innovation and technology management [6].

The participation of entrepreneurs in industrial globalization created an opportunity for developing countries to utilize their resources to build a knowledge-based economy [22]. The significance of knowledge, innovation and technology management on entrepreneurs at the national level, especially among developing countries, allows them to reconfigure domestic R&D [23]. At the minimum level, entrepreneurs in the majority of developing countries are looking to adopt dynamic capabilities for R&D to align with global industrial demand [23]. Since the significance of entrepreneurship is considered a primary driver of knowledge and innovation for economic growth [24]. Knowledge-based entrepreneurship allows developing countries to enhance their commercial R&D capacities at the national level by enabling aggressive national innovation policies [24]. Therefore, the role of entrepreneurs in extending the commercial aspect of the knowledge-based economy (KBE) depends on encouraging individuals to shape their skills, learning attitudes and behaviors to understand the global demand [24]. Such practices allow entrepreneurs to expose and establish capabilities related to knowledge, innovation and technology management at individual, organizational and global levels [24]. Ultimately, such exploration allows them to confront the peculiarities of dynamic globalization on economic and technological fronts.

KNIT management as a supporting management discipline has been an under-researched phenomenon for a very long time [25] and researchers in the field of R&D have explored numerous aspects of this phenomenon [26,27]. The existing literature suggests a number of studies that have been carried out to study the significance of KNIT management capabilities under individual capacities associated with R&D as a core management discipline [28–33]. The impact of KNIT management capabilities on R&D in public organizations was considerably lower in developing countries [34,35].

Some prior studies draw a relationship between knowledge management and R&D, for example Park and Kim [36] suggested that knowledge management processes can be considered as tool for R&D activities in translating information to new products and processes. Dingyong et al. [37] illustrate that knowledge management capability is a core strength for those organizations that are dealing with R&D projects. Similarly, Lilleoere and Holme Hansen [38] explore the impact of knowledge sharing as a core KM process capability on R&D employees in reducing he knowledge barriers and emphasizing a value of synergism. Similarly, capabilities related to innovation management are recognized as a crucial element of economic strength for various developing countries. The national innovation mechanism is quite diverse; every country has different innovation management criteria for dealing with R&D at a national level [39]. According to Lundvall et al. [14], a national innovation mechanism considers open, dynamic and complex innovation management capabilities as a tool for Interorganizational/intra organizational affiliation. Such affiliation justifies the direction of the innovation. The experience-based learning mechanism in this system creates capabilities.

Technology management is also considered a critical component in developing Science, Technology and Innovation (STI) policies in various developing countries [14,39]. Since the policies regarding Science and technology have a different specification, for instance, different countries have different technology management standards for developing their R&D strength [40]. Bolukbas and Guneri [41] evaluated a framework for examining dimensions of technology competency based upon the efficient utilization of technology management capabilities to develop effective R&D at a national level. Wu and Wu [42] discuss the relationship between technology management capability and independent innovation under R&D and identify the relationship between technology management and technological capabilities to upgrade existing R&D at a national level.

Ang and Chai [43] developed a framework that is used to address defense R&D investment and optional theory; the basic concept behind their work is based on the technology management literature with a prime emphasis on developing technological capability for the indigenous defense industry. The prior researches draw insufficient perspective regarding relationship in between capabilities that belongs to knowledge, innovation and technology management with their significant influence on R&D. Prior studies were mostly highlights relationship among all three supporting management discipline at individual level somehow draw less influence on R&D in case of futuristic or philosophical context. Therefore, to address this gap in the literature, this research offers a big data approach that allows researchers to classify capabilities related to knowledge, innovation and technology management based on mapping the resources that drive these capabilities. Also, within a philosophical context, this research allows a pragmatic approach that provides an insightful and rich context in which to address the challenges associated with R&D and practices. Such research paradigm is not limited to questions of how knowledge claims are validated, but also explores alternative orientations.

The conceptual model in this research, based on the modification of theoretical evidence that highlights the description with a relevant clarification of the vital conditions, is shown in Figure 1 [10,44].

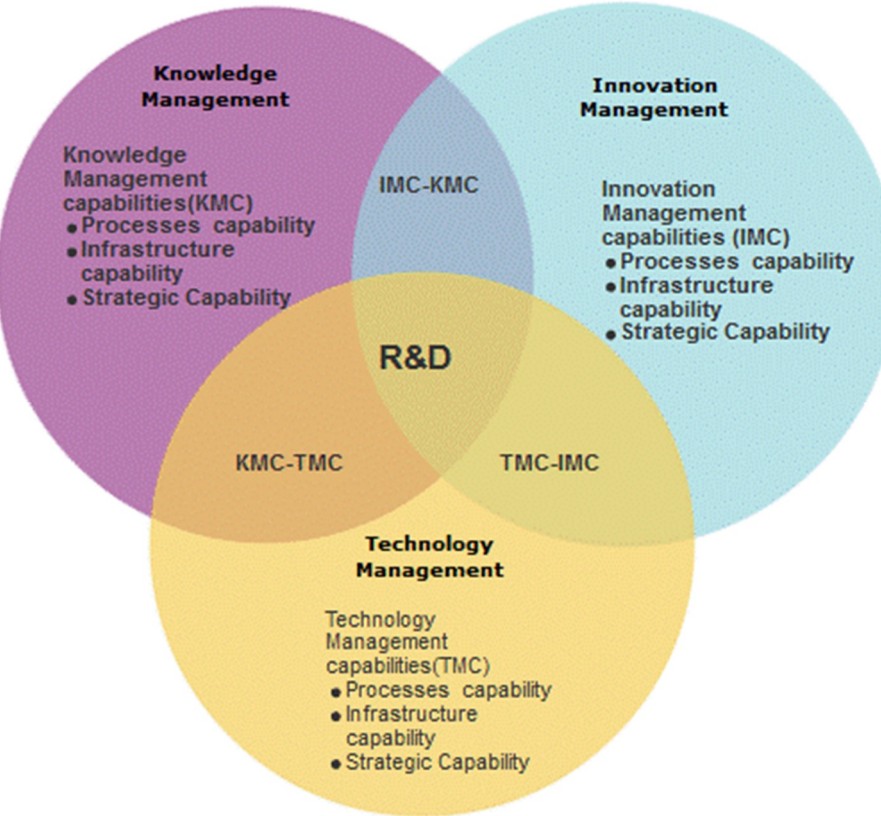

**Figure 1.** Boundaries among innovation, technology and knowledge Management. Source: [10,44].

The conceptual theory allows the researcher to see the specific factors that are crucial in supporting a hypothetical argument. Furthermore, the conceptual Model consists of particular dimensions that justify the critical conditions and are conceived as an imperative for estimating a logical interpretation for developing practical relevancy. The conceptual model suggested in this research assesses the potential capabilities that directly influence generic R&D characteristics. Since, author is aware of the reality that the significant output during the assessment depends upon three influential factors: technology, knowledge and innovation management with unusual interrelation conditions, which were also observed during the systematic review of the literature.

## 3. Methods

Bibliometric interpretations based on PRISMA with a co-word analysis technique were applied in this research. Such extensive research helps the author to investigate the existing studies, gauging the number of published research articles within the domain of specific topics. The extraction of some of the emerging dimensions depends upon reliable sources. In the case of this research, the Scopus database allows the extraction of some of the emerging themes within the scope of a given research domain. In conventional bibliometric techniques, such as author and journal co-citation, exploration is generally based on the assessment of citations that are included in research articles. While this type of extensive analysis provides interesting outcomes, it does not drive an instant picture of the authentic research area content that is compatible with the literature. Co-word analysis counts and assesses the co-occurrence of keywords in a given research topic in a specified research area [45]. Co-word analysis condenses huge data sets into specific visualized patterns that preserve the crucial information enclosed in the data. This analysis depends on the word characteristics, which are considered significant representatives of emerging scientific concepts, creative ideas and new knowledge [46]. In this research, the author followed the PRISMA checklist in order to extract the data as shown in Table 1 [47,48]

**Table 1.** Checklist of items to include when reporting a systematic review (with or without meta-analysis).

| Section | Items | PRISMA | Description |
|---|---|---|---|
| Method | Protocol registration | √ | The systemic review unable for Protocol registration on the PROSPERO database because it comprehensively belongs to R&D capabilities directly and is not connected to any medical or Health-related product. |
| | Eligibility criteria | √ | The initial search uses a comprehensive text mining technique to apply smart keywords in addition to several different configurations. The studies that are rejected under the criteria include the following: (1) research literature available before 1990; (2) the extracted article did not contain the specified keywords; and (3) new opportunities among the three sets of capabilities as catalysts that have an influential impact on R&D. Two neutral research experts screened this research for the applicability of the bibliometric technique and tile and abstract the full description. |
| | Study characteristics | √ | Study design: The systematic review includes all the relevant study designs. The studies that qualified for inclusion included Journal articles, working paper editorial reviews, short surveys, commentaries and Technical notes. Populations: Overall, 2674 studies were included in the data synthesis. *Interventions*: The research literature available before 1990 is excluded. Comparators: All stakeholders that engage in R&D activities were eligible for inclusion. Outcomes: The outcome of bibliometric visualization patterns among knowledge, innovation and technological capabilities after extensive searching string applied on "Scopus" to reclaim all the significant studies related to all three set of capabilities that influence on R&D. Timing: The analysis highlighted most periods from 1990 to 2018. Setting: There were no location restrictions. |
| | Report characteristics | √ | *Language*: Non-English language studies were eligible for inclusion *Publication type: unpublished and published article were identified and with respect to source tile while unpublished studies may have been less likely to satisfy the literature outcome.* |
| | Information sources | √ | Scopus was selected as the major database source for the extensive search over the 1990 to 2018 period. |

**Table 1.** *Cont.*

| Section | Items | PRISMA | Description |
|---|---|---|---|
| | Search strategy | √ | The following criteria were used in this study to search the literature: (1) comprehensive literature review with popular exposure of Knowledge management capabilities typology (ies) or taxonomy (ies), Innovation management capabilities, and technology management capabilities; and (2) a literature review reporting the R&D management typology (ies) or taxonomy (ies). The medium of understanding scientific publication language is English. The overlapping research publications were also analyzed and excluded after a comprehensive review. |
| | Study selection | √ | Overall, the studies that were found were downloaded into Microsoft Excel in the CSV file (Comma separated value) format from the Scopus database from the 1990 to 2018 period. A panel of three neutral researchers independently evaluated the results for overlapping studies by contrasting the tile, the author name and the study abstract. If the studies were replicated, they were screened out by analyzing the full research manuscript to identify if they were identical articles; if so, one would be excluded. vosviewer software used to construct the bibliometric pattern was based on the large quantity of data downloaded from Scopus database. |
| | Selection process | √ | The Sci2 tool is applied for selection process which currently uses the Kleinberg's burst detection algorithm [49], that assesses unexpected increases in the occurrence of words. The basic mechanism behind the algorithm allows a probabilistic estimation that responds when there is an increasing occurrence of individual words. State switches correspond to the approximate time at which the occurrence of words significantly adjusts. The studies that were used to screen for inclusion depend on the co-occurrence key-phrases that appear in other studies that have also been extensively analyzed. All the relevant keywords can be screened either from the author-supplied keyword or extracted from the title and the abstract of the research publication. Any disagreements regarding the inclusion processes was noted and were discussed with an experts to determine whether a research article should be included. Any causes for exclusion were recorded. |
| | Data collection process | √ | Data extraction was approved by an individual expert with independent review assisted by a supervisor who verified the data mining instrument accuracy |
| | Data item | √ | Data was extracted from the eligible studies and summarized in Table: 6, 12, and 18. After vigilant assessment, twenty eight research studies were selected; these revealed three sets of criteria: (1) process capabilities, (2) infrastructure capabilities and (3) strategic capabilities. Data (dimensions, criteria and sub-criteria) included specific study distinctiveness; most significantly, on the degree to which the study theme and preliminaries reveals the nature of each criteria. |
| | Risk of bias in individual studies | √ | As the all selected studies have already been published, this section is not relevant to our review. |
| | Summary measures | √ | A systematic descriptive analysis will be carried out in order of assess the degree to which studies meet the relevant criteria. Any missing items or data will be addresses in the result with detailed argument. |
| | Risk of bias across studies | √ | In this systematic review we did not gauge the cumulative quality of the studies; this is not requirement in our review. |

## 4. Results

### 4.1. Assessing Knowledge Management Capabilities

This study explores new opportunities in knowledge management capabilities as catalysts that have an influential impact on R&D activities. All the assessment and validation led to a new practical evolution in which the retained R&D capacity remains adaptable during any condition as shown in Figure 2.

We acknowledged 7892 relevant articles by systemically searching on the Scopus Database. After removing research articles that did not fulfill the eligibility criteria based on PRISMA, a total of 1040 articles were recognized. Articles from the period 1990–2018 were analyzed, with the number of research articles with author supplied keywords was 512 and the number of research studies without keywords was 528. Research articles were analyzed on the basis of Tile, Keywords and Abstract. We exclude overall 6852 articles based on eligibility criteria. The complete visual pattern is presented in Figure 3.

The search string applied on the Scopus database to retrieve all the significant studies related to Knowledge management capabilities that influence R&D. The following typology configuration was applied to the Scopus search engine: Searched for article: "Knowledge and management" OR "Knowledge organization capabilities" OR "Knowledge capabilities" OR "Knowledge capacity" OR "Knowledge Management in R&D" OR "Knowledge Management" OR "Knowledge Management" OR "Knowledge Capabilities" OR "Knowledge ability" OR "Knowledge ability" OR "K.M capabilities" OR "K.M" OR "K.M "All the probable keywords relevant to Knowledge management Capabilities (K.M capabilities) were taken into account during the systematic searching query. The studies that were found were downloaded into Microsoft Excel in a CSV file (Comma separated value) format from the 1990 to 2018 period.

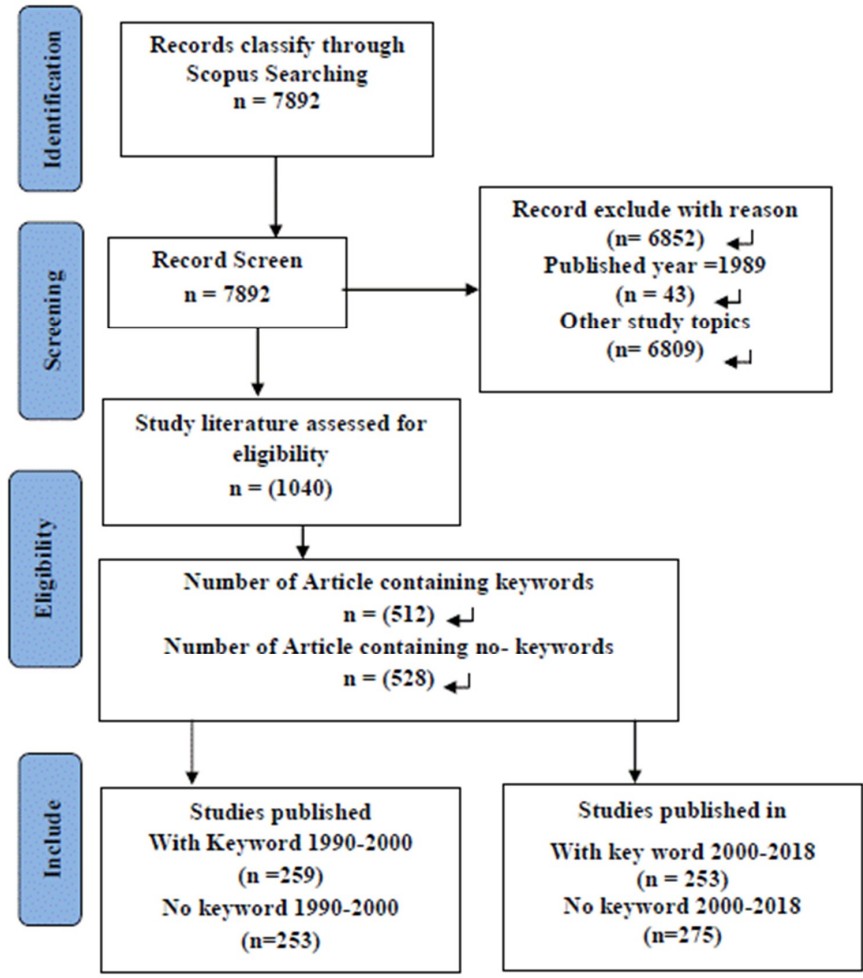

**Figure 2.** PRISMA Flow diagram for Knowledge Management Capabilities during 1990–2018.

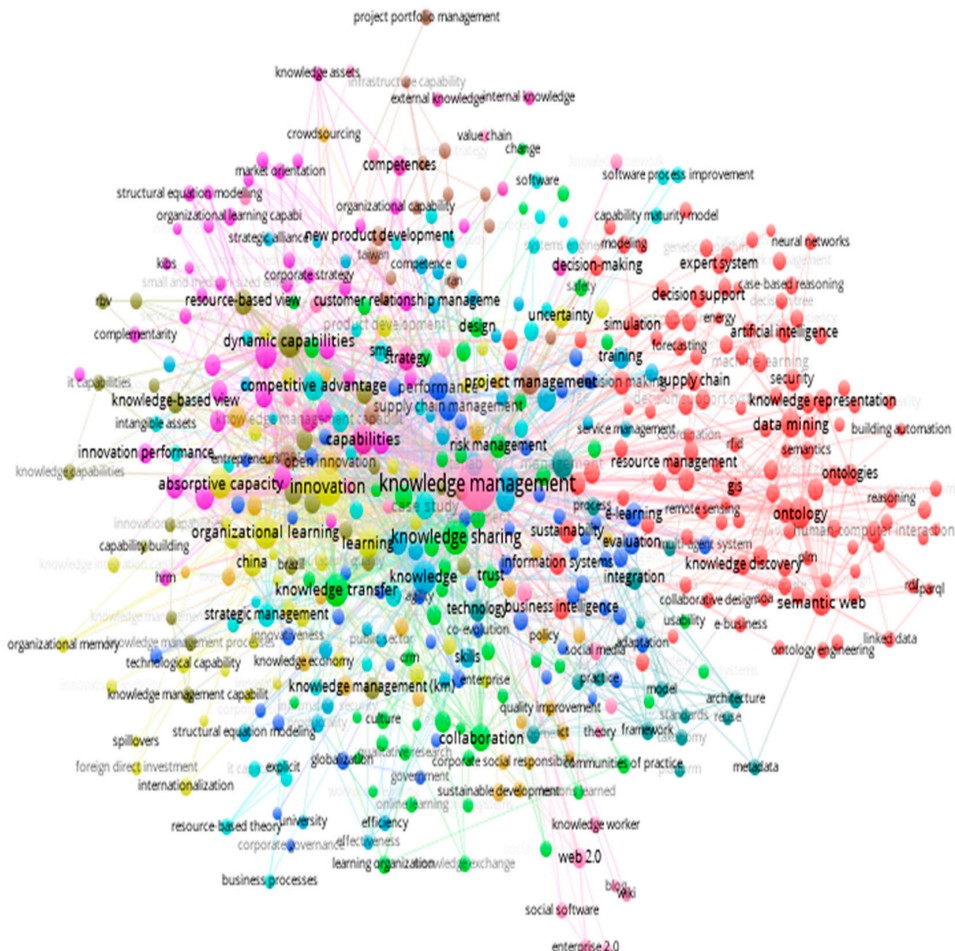

**Figure 3.** The term maps from the 1990 to 2018 period.

Based on multi-dimensional scaling (MDS) techniques, the vosviewer at the initial level mapped numerous inflow and outflow edges between the nodes. The primary goal of MDS (Multidimensional Scaling) is to navigate items in a low dimensional space in a way such that the gap between any two texts represents the same meaning or similarity as precisely as possible. The higher the strength between two texts, the smaller gap in between them.

The first aspects are simply known as "Label" and are considered the most crucial output aspects within the vosviewer system, which represents the comprehensive list of keywords. These keywords have a number of recurrences in the publications. Some key phrases are normal words and have common meanings which are associated with every research publication related to the research scope. Some of these words frequently occur, for instance words like 'articles,' 'conference,' 'experiments' and so forth. However, these keywords do have common meanings that reflect the particular research theme or specific area of study. Thus, it is important to identify which specific words can provide high impact to a research study. The second most significant aspect of the VOS viewer system is the representation of words in cluster form. The initial concept of cluster according to the Merriam-Webster dictionary is the representation of a number of related or similar items that appear together—for instance two or more excessive consonants or vowels within a fragment of speech. Therefore, in the vosviewer output, there is a list of the groups of keywords with respect to close or similar definitions to one another. vosviewer will assist researchers in determining the number of clusters with similar research themes. The higher the number of clusters, the bigger the group of words that will appear collectively together. Within these clusters, researchers can determine the significant research themes or literature to be used for systematic review. The third aspect reflected in the vosviewer output is

a strong connection between these words which is also known as 'link.' A simple word link is used to signify the total number of words that have a similar meaning. Last but not least, the occurrences are a comprehensive frequency of recurrence of words in the data. The default number of minimum occurrences that has been suggested for vosviewer is 5 repetitions, which means that one specific word such as 'system' can recur about five times in an entire publication.

In simple words, the "node" represents the strength of key phrases, while the "edges" represent the occurrence among two entities. For this research, a distance-oriented visualization pattern that was adapted to analyze the occurrence and co-occurrence of keywords has been illustrated as a graph–oriented visualization. A panel of three neutral researchers independently evaluated the results for overlapping studies by contrasting the tile, the author name and the study abstract. If the studies were replicated, they were screened out by analyzing the full research manuscript to identify whether they were identical articles; if so, one would be excluded. Some of the highest occurrences of key phrases are shown in Table 2.

The 1st cluster is comprised of the majority of key phrases that are related to computers and artificial intelligence. The 2nd cluster's majority of key phrases belong to studies that illustrate the general perspective of organizational learning including current and future trends based on existing organizational leadership confrontation of any future challenges related to market demand.

The 3rd cluster represents keywords that belong to some of the articles that utilized knowledge management as a tool for new product development undermining the relevance of some capabilities related to extensive R&D. Within the same clusters there are some other trends of studies on social networking. In the 4th cluster, the majority of key phrases represent study trends related to capability that used knowledge management as an instrument for community of practice. In this cluster, the majority of studies focused more on utilizing knowledge management tools for social purposes. The 5th cluster represents the key phrase that shows trends in which the majority of studies utilize knowledge management as a tool to sustain Total Quality Management (TQM) in order to get maximum competitive advantage. In the same cluster, few studies represented technology transfer as a tool to apply innovation by improving knowledge management for the banking sector. The 6th cluster shows that the majority of studies' trends are related to process capability related to knowledge management in the R&D context. Cluster 7 represents the majority of studies that show overlapping trends related to infrastructure capability related to knowledge management in the R&D context.

The 8th cluster shows some studies' trends belong to strategic capability related to knowledge management in the R&D context. In a similar fashion, the 9th cluster shows studies with trends of firms putting more emphasis on utilizing knowledge management tools for social and external networking in order to sustain their progressive value chain. The 10th cluster is used to represent the studies' trends where the majority of firms put more emphasis on utilizing knowledge management as a tool to enhance their organizational performance. The 11th cluster similarly puts more focus on business performance for strategic means. In clusters 12 and 13 the study trends put more emphasis on applying knowledge management as a tool for infrastructure and environmental development in order to develop more strategic alliances.

**Table 2.** Keywords with Highest Occurrence (KM Capabilities).

| Cluster | Keywords with Highest Occurrence | Cluster | Keywords with Highest Occurrence | Cluster | Keywords with Highest Occurrence |
|---|---|---|---|---|---|
| Cluster 1 | Data mining, Artificial intelligence, Business Intelligence, Business process management, cloud computing, Data mining, Database, Decision making, Decision support system, Information retrieved, Information management | Cluster 5 | Competitiveness, banking, Critical success factor, Information technology, Technology Transfer, Total Quality, Six sigma | Cluster 9 | Knowledge network, leadership, Learning organization, value chain, value creation, social software, social network system, system review |
| Cluster 2 | Assessment, Communication, Disasters management, E-learning, Evaluation, Higher Education, HRM, Knowledge Management, Leadership, Organization learning | Cluster 6 | Knowledge acquisition, Knowledge application, Knowledge conversion, knowledge creation, knowledge generation, knowledge integration | Cluster 10 | Organizational performance, Organizational capability Organizational design, organizational learning, Dynamic capability, knowledge based view |
| Cluster 3 | Case study, Competitive Advantages, Information, Knowledge Management, New Product Development, SMEs, Social Networks | Cluster 7 | Innovation Management, Intellectual capital, knowledge economy, Open innovation, Organization innovation, strategic management, R&D management, Process innovation, organizational culture, Organizational learning; Culture, IT, Community of Practice, Technology, Structure; People, Contribution of skills & expertise | Cluster 11 | Business performance, business strategy, Strategic alliance, project portfolio management, Quality, SMEs |
| Cluster 4 | Capability, Collaboration, Community of Practice, Creativity, Education, knowledge, Learning Training, Nurse | Cluster 8 | External Knowledge, Internal knowledge, implicit, Explicit, Technology strateg, strategic alliance, combination, internalization, and socialization | Cluster 12 | Architecture, development, environment, framework, infrastructure, integration, practices |
| Cluster 13 | Innovativeness, Hotel industry, Agility, corporate social response, ICT, Alliance | | | | |

A total of 15,534 authors supplied keywords; however, only 692 met the minimum occurrence threshold value; therefore, the 692 keywords were split into 13 clusters. A complete descriptive analysis produce by VOS viewer software was developed and is shown in Table 3.

**Table 3.** Descriptive analysis of each knowledge management capabilities study concentrated on research and development (R&D) from 1990–2018.

| Years | 1990–2000 | 2001–2018 |
|---|---|---|
| Total Paper | 512 | 528 |
| Minimum no of keywords | 107 | 2243 |
| Minimum occurrence | 5 | 5 |
| Minimum Threshold | 13 | 112 |
| Highest total link strength | 245 | 232 |
| Highest occurrence | 28 | 165 |

### 4.2. Emerging and Disappearing Themes (Burst Detection)

In order to get a more specified outcome, the extracted data was used for further analysis by applying a burst detection technique to the extracted dataset in order to explore emerging and faded themes. After applying burst detection techniques, 17 emerging and fading themes appeared—both title and author supplied keywords with respect to the time frame are shown in Table 4. The basic mechanism behind the algorithm allows a probabilistic estimation that responds when there is an increasing occurrence of individual words. State switches correspond to the approximate time at which the occurrence of words significantly adjusts. The studies that were used to screen for inclusion depend on the co-occurrence of key-phrases that appear in other studies that have also been extensively analyzed.

**Table 4.** Keywords of emergent and fading subjects.

| Latest Bursting and Disappearing Topics | | | | | | | | | | | |
|---|---|---|---|---|---|---|---|---|---|---|---|
| **In Author Supplied Keywords** | | | | | | **In the Titles** | | | | | |
| Word | Level | Weight | Length | Start | End | Word | Level | Weight | Length | Start | End |
| combin | 1 | 3.689 | 1 | 2016 | | volum | 1 | 8.146203 | 3 | 2011 | 2013 |
| chang | 1 | 3.591 | 2 | 2011 | 2012 | technolog | 1 | 3.82957 | 11 | 1992 | 2002 |
| explicit | 1 | 6.2053 | 9 | 1998 | 2006 | sustain | 1 | 3.57236 | 2 | 2009 | 2010 |
| implicit | 1 | 7.290 | 9 | 1998 | 2006 | 2012 | 1 | 4.380297 | 1 | 2012 | 2012 |
| extern | 1 | 6.5064 | 7 | 1997 | 2003 | 2013 | 1 | 6.13927 | 1 | 2013 | 2013 |
| extern | 1 | 8.4390 | 1 | 2016 | | America | 1 | 5.614195 | 2 | 2012 | 2013 |
| collabor | 1 | 7.1013 | 8 | 1994 | 2001 | servic | 1 | 6.476224 | 2 | 2010 | 2011 |
| social | 1 | 5.5239 | 1 | 2016 | | amci | 1 | 5.614195 | 2 | 2012 | 2013 |
| inform | 1 | 3.4616 | 1 | 2013 | 2013 | confer | 1 | 8.482029 | 2 | 2012 | 2013 |
| intern | 1 | 7.0512 | 7 | 1997 | 2003 | inform | 1 | 10.58485 | 2 | 2012 | 2013 |
| intern | 1 | 4.4818 | 1 | 2016 | | empir | 1 | 3.918114 | 7 | 2002 | 2008 |
| base | 1 | 3.6587 | 8 | 1996 | 2003 | ici | 1 | 3.476513 | 3 | 2011 | 2013 |
| innov | 1 | 4.3385 | 1 | 2011 | 2011 | 18th | 1 | 3.748391 | 1 | 2012 | 2012 |
| technolo | 1 | 4.5136 | 11 | 1997 | 2007 | research | 1 | 1.908279 | 1 | 2012 | |
| human | 1 | 8.1746 | 27 | 1976 | 2002 | book | 1 | 1.479245 | 1 | 2010 | 2010 |
| perform | 1 | 9.0109 | 27 | 1976 | 2002 | mechan | 1 | 1.652 | 1 | 2010 | 2010 |
| process | 1 | 5.8698 | 3 | 2014 | | patient | 1 | 1.54516 | 1 | 2012 | |

Currently, in the field of KMC (knowledge management capabilities), the most crucial key phrase themes used to influence the process, infrastructure and strategic domains within R&D, are represented by prospective keywords which include: combination (2016-Active), internalization (2016-Active), and socialization (2016-present). These keywords represent the probable research themes that are very active in the current research pattern; conversely, there are faded themes that are not significant and that are less followed in contemporary research trends. The following keywords have not been included in either the author's supplied keyword list or in the research titles: implicit (1998–2016), Explicit (1998–2006), external (1997–2003), internal (1997–2003), Performance (1976–2002), base (1996–2003),

system (2012–2013), Human (1976–2002), and collaboration (1994–2001); these have appeared less in contemporary research. After figuring out the emerging and faded themes the author extracted some of the emerging trends related to the scope of the study as shown in Figure 4.

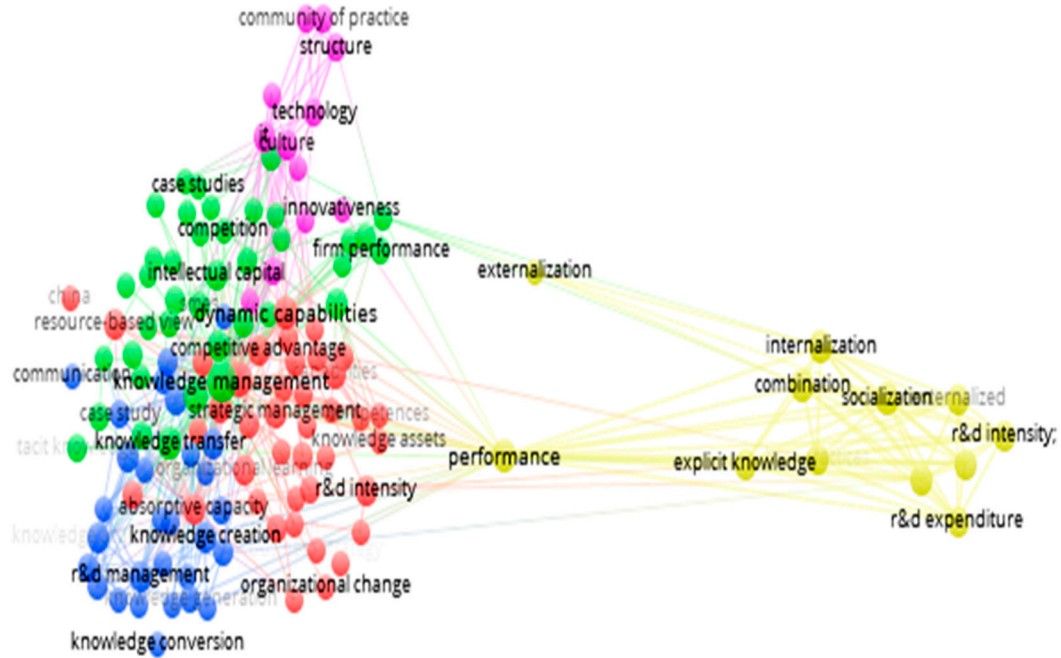

**Figure 4.** VOS viewer Pattern of Knowledge Management Capabilities.

It is quite appealing that, after extensive analysis, in cluster 3 (Blue in color) the nature occurrence-keyword is closer to knowledge management capabilities. This cluster is more aligned to reflecting capabilities closer to the processes perspective related to knowledge management. Some of keywords are represented as: Knowledge sharing, Joint scene-making, Knowledge Implementation, Knowledge Transfer, Knowledge creation, Knowledge generation, Knowledge protection, Knowledge Acquisition, Knowledge Utilization. While, cluster 4 (yellow in color), expanded with an in-depth focus on occurrence-keywords, reflects a more strategic aspect related to knowledge management capabilities along with internal and external organizational dimensions. Some of key trends are represents as: External knowledge sourcing; internal knowledge sourcing; explicit knowledge; joint learning internal collaboration; joint learning external collaboration; Externalized; Internalization; Combination; Socialization; R&D expenditure. Finally, cluster 5 is (Pink in color) the pattern of occurrence keywords that directly referred to infrastructure that arranges comprehensive knowledge infrastructure interfaces with firms' existing culture and structure. Some of the trends are represented as Organizational learning; Culture; IT; Community of Practice; Technology; Structure; People; Contribution of skills & expertise. Occurrence-keywords help to identify the traces of relevant studies through which knowledge management capabilities, that have influential enablers, may recognize the potential driving factors behind the dominance of a knowledge management capability's influential impact on R&D, as shown in Table 5.

This study explores new opportunities for knowledge management capabilities as a catalyst that shares influence with R&D. Not only that, it has also led researchers to understand the assessment and validation of a new practical evolution that retains R&D competitiveness. After careful consideration, there are some studies emerging that closely relate to the topics shown in Table 6.

**Table 5.** Knowledge management (KM) capabilities in the R&D context.

| Authors | Process | KM Infrastructure | Strategy |
|---|---|---|---|
| Denicolai et al. [49] | N/A | N/A | External Knowledge source Internal Knowledge source |
| Bäck and Kohtamäki [50] | Knowledge sharing Joint scene-making Knowledge Implementation | N/A | Joint learning: Internal collaboration; External collaboration |
| Žemaitis [51] | Knowledge Transfer | N/A | Tacit Knowledge (Personalization); Explicit Knowledge (Codification) |
| Potgieter et al. [52] | Knowledge sharing; Knowledge creation | Organizational learning; Culture; Structure | K.M strategy: Personalization (Human-oriented) |
| He-jiang [53] | Knowledge Acquisition | N/A | N/A |
| Zammit and Woodman [54] | N/A | N/A | Codification; Personalization |
| Camelo-Ordaz et al. [55] | Knowledge Sharing; Organization commitment | N/A | HRM Practices; Performance |
| Satyanarayan and Gideon [56] | Knowledge generation; Knowledge protection | Culture; IT; Community of Practice | Codification |
| Jain et al. [57] | Intellectual knowledge portfolio | Contribution of skills & expertise; Novelty & uniqueness of innovation; Role of leadership innovation & supports | R&D expenditure; Success rate of R&D products; R&D intensity |
| Liao et al. [58] | Knowledge Creation; Knowledge Sharing; Knowledge Utilization | Technology; Structure; Culture; People | N/A |

**Table 6.** Internal Determinates of K.M capabilities in the R&D context.

| | | Enablers | References |
|---|---|---|---|
| **Knowledge Management Capability** | **Process** | Knowledge sharing | [59–65] |
| | | Joint scenes Making | [59,66–69] |
| | | Knowledge Implementation | [70–72] |
| | | knowledge Creation | [58,73–76] |
| | | Affective commitment | [55,77–82] |
| | | knowledge Utilization | [52,83–85] |
| | | knowledge Transfer | [86–88] |
| | | Knowledge Protection | [56,89,90] |
| | | knowledge Acquisition | [56,91,92] |
| | | Intellectual knowledge portfolio | [57,93–95] |
| **Knowledge Management Capability** | **Infrastructure** | Organization Learning | [96–100] |
| | | Culture | [56,91–101] |
| | | Structure | [58–107] |
| | | Technology | [56,58,82,108,109] |
| | | People | [58,105,109,110] |
| | | Community of Practice | [56,111–113] |
| | | IT | [114–117] |
| | | Contribution of skills & expertise | [57,118,119] |
| | | Novelty & uniqueness of innovation | [57,120,121] |
| | | Role of leadership innovation & supports | [122–124] |
| **Knowledge Management Capability** | **Strategies** | External Knowledge Source | [125–132] |
| | | Internal Knowledge Source | [125,133–136] |
| | | Joint internal Collaboration | [50–137] |
| | | Joint External Collaboration | [50–145] |
| | | HRM | [55,81,146] |
| | | Innovation Performance | [69,136–153] |
| | | Explicit | [51,52,56,154] |
| | | Tacit Knowledge | [107,155,156] |
| | | Codification | [51,52,54] |
| | | Personalization | [51,52,54] |
| | | R&D expenditure | [69,119–163] |
| | | Success rate of R&D products | [57,62,95,164] |
| | | R&D intensity | [156,165,166] |

### 4.3. Assessing Innovation Management Capabilities

Innovation management capabilities strongly referred to a firm's core ability to manage R&D for new product development [167]. In simple words, innovation management capability is not only sufficient for radical innovation at a governmental level, but it also promotes science and technology to enhance R&D competitiveness in order to create new innovative products. For example, accommodating accessibility to internal and external collaboration, encouraging the relevant environment for social exchanges, and strong research support mechanisms [168].

Innovation has been recommended as major contributors to public sector firms' drive overall innovation mechanisms at a governmental level [169–171]. This study suggested a new opening in innovation management capabilities as a potential booster to enhance R&D competitiveness, with all the estimation and justification that leads to new practical progression.

We accepted 6769 relevant articles by systemically searching the Scopus Database. After removing research articles that did not fulfill the eligibility criteria based on PRISMA—already discussed in chapter 3-a total of 972 articles were recognized. Research articles with author-supplied keywords (n = 619) from the 1990–2018 period and research studies without keywords (n = 353) from the 1990–2018 period were analyzed. The research articles were analyzed based on Tiles, Keywords and Abstracts. Because of the characteristics described in the eligibility criteria, we excluded a total of 5797 articles. The comprehensive representation of the record exclusion at each stage is shown in the PRISMA flow diagram in Figure 5. The outcome of the bibliometric visualization analysis of innovation management capabilities from the sequential point of view is driven by the smart configuration of key phrases with a unique typological pattern used to apply during the advance searching string as we discussed earlier. The analysis highlighted the 1990 to 2018 period, as shown in Figure 6.

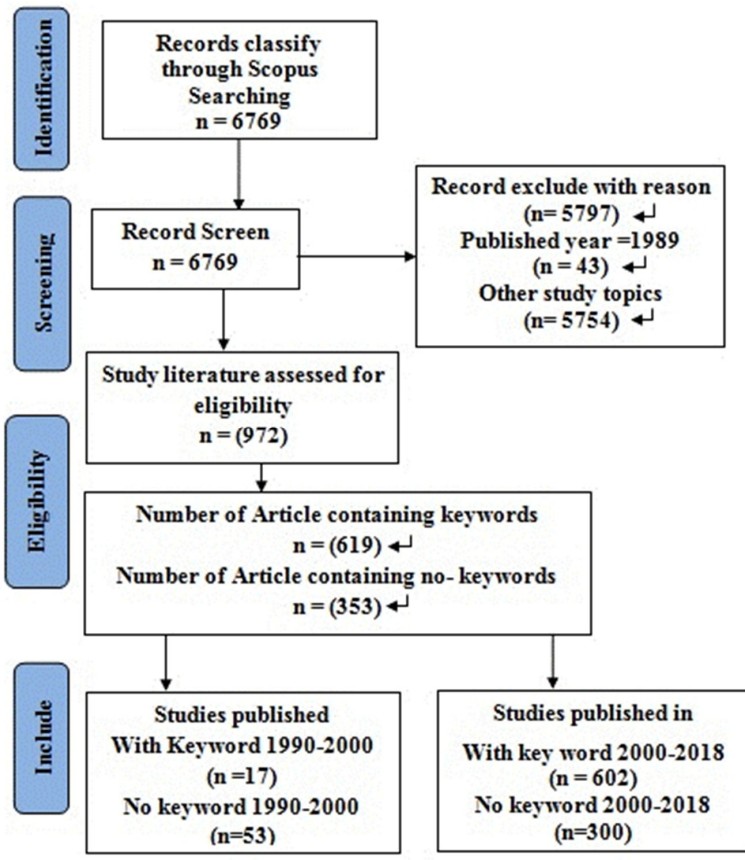

**Figure 5.** PRISMA flow diagram for Innovation Management Capabilities during 1990–2018.

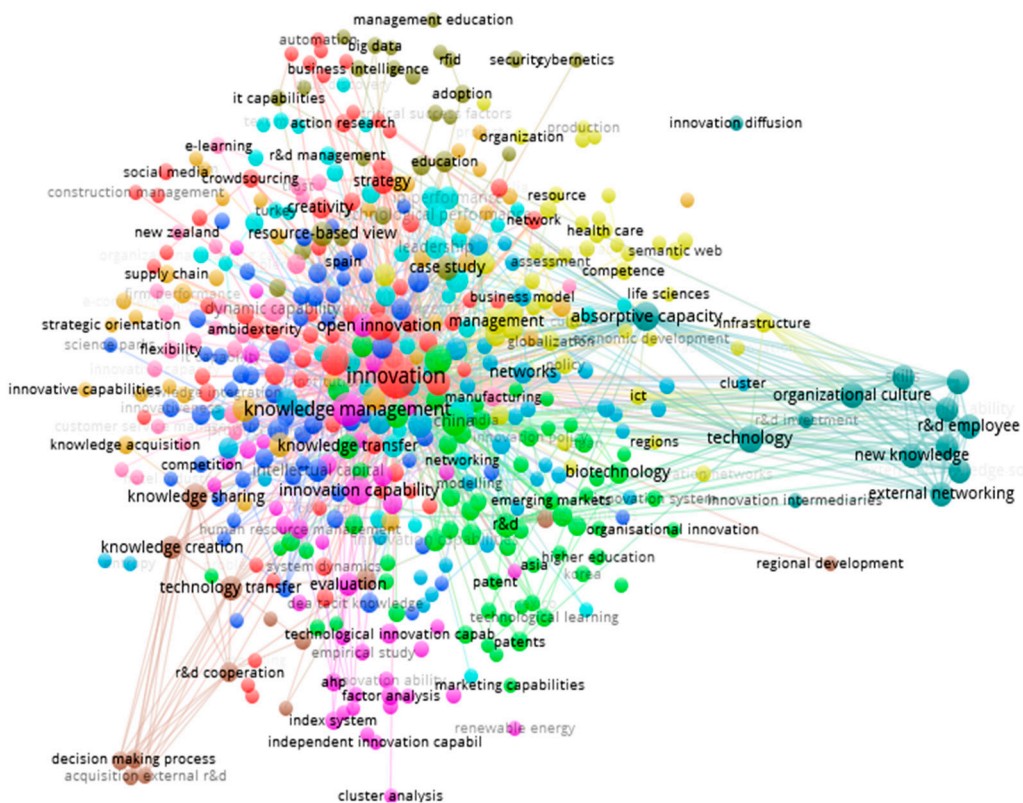

**Figure 6.** Vosviewer Pattern of Innovation Management Capabilities.

In the case of innovation management capability the following typology configuration was applied to the Scopus search engine: "Innovation Management capacity" OR "Innovation organization" OR "Innovation management" OR "Innovation and Research" OR "Innovation capabilities SMEs" OR "I.M in MEs" OR "Innovation management and (R&D)" OR "Innovation Capability" OR "Innovation & service" OR "Innovation management capability & research and development" OR "Innovation capabilities" OR "Innovation & Capabilities" OR "Innovation for research organization". All the probable keywords relevant to innovation management capabilities (IM capabilities) were taken into account during the systemic searching query. Some of the highest occurrences of keyphrases are shown in Table 7.

**Table 7.** Keywords with Highest Occurrence (IM capabilities).

| Cluster | Keywords with Highest Occurrence | Cluster | Keywords with Highest Occurrence | Cluster | Keywords with Highest Occurrence |
|---|---|---|---|---|---|
| Cluster 1 | Business process management, capabilities, corporate culture, corporate entrepreneurship, dynamic capability, knowledge economy, knowledge integration, open innovation, organizational capability, organizational change, performance | Cluster 5 | Corporate governance, corporate social, responsibility, networking, organizational learning, R&D management, technological innovation capabilities, collaborative innovation, green innovation | Cluster 9 | Cloud computing, cybernetics, big data, information system, IT innovation, IT capability |
| Cluster 2 | Innovation capability, innovation management, competitiveness, intellectual property, knowledge transfer, knowledge management, management capabilities | Cluster 6 | Technological performance, technology trends initiative, strategic management, strategic planning | Cluster 10 | Manufacturing strategy, manufacturing, strategic flexibility, flexibility, organizational development, communication, competencies |
| Cluster 3 | Diffusion innovation, exploitation, innovativeness, information technology, knowledge management, process management, product development, resource management, product innovation, total quality management, diffusion innovation | Cluster 7 | Business environment, Business development, commercialization, portfolio management, total innovation management, uncertainty and venture | Cluster 11 | Decision making process, acquisition external R&D, acquisition internal R&D, knowledge sharing, knowledge creation, R&D cooperation, technology transfer, in bound open innovation ability |
| Cluster 4 | Innovation process, policy, radial innovation, economic development, environmental management, globalization, green innovation | Cluster 8 | Knowledge acquisition, knowledge innovation, organizational capabilities, regional innovation mechanism, resource and capabilities, supply chain, supply chain integration, strategic orientation | Cluster 12 | Absorptive capability, R&D investment, organizational culture, new knowledge, external knowledge source, formulization ability |

The 1st cluster comprises key phrases that represent the studies' trends related to corporate culture and corporate entrepreneurship which is necessary for the knowledge economy. With the same clusters there were a few other trends also emerging that represent dynamic capabilities related to innovation management, used as a potential tool for organizational change and performance. The 2nd cluster includes a majority of key phrases related to the overlapping studies trends of capabilities related to knowledge and innovation management. A majority of these studies illustrate the general narrative regarding these capabilities getting maximum competitive advantage by improving their innovational ability to enhance their intellectual property.

The 3rd cluster comprises a majority of key phrases about utilizing innovation management as a tool for innovation diffusion in order to improve firms' existing process and product development requirements. There are a few other study trends also emerging within the same cluster that represent the general narrative of innovation management for developing innovativeness in the information technology (IT) industry. The 4th cluster represents research trends towards developing innovation policy as a tool for economic development in developing countries. Within the same clusters there were a few other research trends also emerging in which a majority of firms developed innovation policy based on environmental friendliness. Similarly, the 5th cluster is about utilizing capabilities related to innovation management for social and corporate governance. Within the same clusters there are a few other emerging trends that represent the extensive utilization of capabilities related to innovation management for effective R&D management and organizational learning.

In a similar fashion, the 6th cluster includes research studies that portray the involvement of innovation management as a tool for measuring technological performance to achieve long term strategic goals. Similarly, the 7th cluster includes study trends about government level firms utilizing innovation management capabilities as tools for commercial purposes allowing firms to respond with respect to the business environment. During the extraction, the majority of these studies placed more emphasis on developing their current innovation portfolio in order to deal with uncertain market demand.

In the 8th cluster, a majority of key phrases were used to represent overlapping concepts related to knowledge and innovation management utilizing their capabilities for sustainable supply chain integration to allow firms to get strategic advantages. In the case of the 9th cluster, a majority of key phrases represent the study trends that are based on utilizing the innovation capabilities for information technology, especially big data and cloud computing. While, the 10th cluster includes the key phrases that represent study trends based on utilizing innovation capability for flexible manufacturing and organizational development. Within the same cluster, organizations were looking more to enhance their capacity and the level of their competencies based on their existing innovational strength. In a similar fashion, the 11th and 12th clusters were more relevant to R&D. In the majority of cases, firms utilizing their process, infrastructure and strategic capabilities were related to innovation management in the R&D context, whether utilizing their decision making resources or spreading their resources strategic corporation. Although 9394 authors supplied keywords, only 480 met the threshold minimum 5; therefore, nearly 9394 author supplied keywords that were traced from the corpus. The 480 keywords were split into 5 clusters. A complete descriptive analysis produced by the vosviewer software was developed and can be shown in Table 8.

**Table 8.** Descriptive Analysis of Each Innovation Management Capabilities Study Concentrated on R&D from 1990–2018.

| Descriptive Analysis 1990–2018 | | |
|---|---|---|
| Years | 1990–2000 | 2001–2018 |
| Total paper | 70 | 902 |
| Minimum no of keywords | 66 | 873 |
| Minimum occurrence | 3 | 5 |
| Minimum threshold | 3 | 26 |
| Highest total link strength | 6 | 219 |
| Highest occurrence | 6 | 51 |

## 4.4. Emerging and Disappearing Themes (Burst Detection)

In order to get a more specified outcome, the extracted data were further analyzed by applying a burst detection technique to the extracted dataset in order to explore emerging and faded themes. After applying burst detection techniques, 27 to 29 emerging and fading themes appeared after exploring; both title and author supplied keywords with respect to time frame are shown in Table 9. The basic mechanism behind the algorithm allows a probabilistic estimation that responds when there is an increasing occurrence of individual words. State switches correspond to the approximate time at which the occurrence of words significantly adjusts.

**Table 9.** Keywords of emergent and fading subjects.

| Latest Emerging and Faded Topics | | | | | | | | | | | |
|---|---|---|---|---|---|---|---|---|---|---|---|
| In the Title | | | | | | Author Supplied Key Words | | | | | |
| Word | Level | Weight | Length | Start | End | Word | Level | Weight | Length | Start | Ed |
| success | 1 | 2.507393 | 2 | 2006 | 2007 | analysi | 1 | 3.411704 | 1 | 2008 | 2008 |
| energi | 1 | 2.524447 | 1 | 2010 |  | R&D | 1 | 6.309655 | 3 | 2003 |  |
| R&D | 1 | 2.676281 | 2 | 2004 |  | product | 1 | 4.712268 | 2 | 2001 | 2002 |
| product | 1 | 6.210285 | 3 | 2001 | 2003 | process | 1 | 3.506707 | 1 | 2005 | 2005 |
| element | 1 | 2.649457 | 1 | 2010 |  | corpor | 1 | 2.736561 | 2 | 2003 | 2004 |
| develop | 1 | 2.925772 | 2 | 2001 |  | univers | 1 | 3.135266 | 1 | 2008 | 2008 |
| small | 1 | 3.206891 | 1 | 2006 | 2006 | custom | 1 | 2.731154 | 1 | 2006 | 2006 |
| organ | 1 | 2.963422 | 4 | 2003 | 2006 | global | 1 | 2.518794 | 2 | 2007 | 2008 |
| 2010 | 1 | 2.649457 | 1 | 2010 |  | disrupt | 1 | 2.436466 | 3 | 2003 | 2005 |
| 2009 | 1 | 3.873894 | 2 | 2009 |  | internet | 1 | 3.35806 | 2 | 2002 | 2003 |
| 2008 | 1 | 2.690328 | 1 | 2008 | 2008 | strategi | 1 | 2.76029 | 3 | 2001 | 2003 |
| 2007 | 1 | 3.388337 | 1 | 2007 | 2007 | research | 1 | 6.745098 | 3 | 2001 | 2003 |
| nuclear | 1 | 2.61664 | 2 | 2003 | 2004 | technolog | 1 | 4.365171 | 1 | 2002 | 2002 |
| health | 1 | 2.464596 | 2 | 2003 | 2004 | mechan | 1 | 3.207627 | 1 | 2010 |  |
| competit | 1 | 2.511899 | 2 | 2001 | 2002 | continu | 1 | 2.479377 | 3 | 2001 | 2003 |
| firm | 1 | 2.841077 | 1 | 2003 | 2003 | compet | 1 | 3.131403 | 2 | 2002 | 2003 |
| agenda | 1 | 2.489341 | 4 | 2001 | 2004 | collabor | 1 | 2.67418 | 3 | 2006 | 2008 |
| share | 1 | 2.810559 | 1 | 2007 | 2007 | R&D | 1 | 2.088961 | 1 | 2012 |  |
| new | 1 | 4.053195 | 3 | 2002 | 2004 | absorpt | 1 | 2.565614 | 1 | 2012 | 2012 |
| tool | 1 | 2.480229 | 3 | 2003 | 2005 | intern | 1 | 2.619999 | 1 | 2012 | 2012 |
| continu | 1 | 2.438552 | 1 | 2008 | 2008 | collabor | 1 | 2.67418 | 3 | 2006 | 2008 |
| integr | 1 | 3.056669 | 1 | 2005 | 2005 | retract | 1 | 6.510619 | 1 | 2010 |  |
| compet | 1 | 2.489125 | 2 | 2002 | 2003 | mechan | 1 | 3.854026 | 1 | 2010 |  |
| key | 1 | 2.679947 | 5 | 2001 | 2005 | evalu | 1 | 7.319149 | 2 | 2009 |  |
| sector | 1 | 2.697859 | 4 | 2003 | 2006 | base | 1 | 2.725118 | 1 | 2010 |  |
| lead | 1 | 3.392705 | 4 | 2002 | 2005 | energi | 1 | 2.524447 | 1 | 2010 |  |
| strategi | 1 | 2.522441 | 1 | 1995 | 1995 | independ | 1 | 2.923765 | 2 | 2009 |  |
|  |  |  |  |  |  | proceed | 1 | 2.469733 | 3 | 2008 |  |
|  |  |  |  |  |  | articl | 1 | 6.996379 | 1 | 2010 |  |

The above themes signify that the studies belong to R&D from the co-occurring keyword perspective, which includes the following: Evaluation (2010-Active), Base (2010-Active), independent (2009-Active), and retract (2010-Active). These themes are the probable research themes that are very active in the current research patterns. Conversely, there are certain faded themes that are not significant and that follow contemporary research trends less; these keywords are not included in the author's supplied keyword list or in the research titles: Development (2001–2002), Competitive

(2001–2002), strategies (1995–1995), integration (2005–2005), and health (2003–2004). These key words splits into 5 clusters that associated with respect to the nature of their characteristics and co-occurrence is shown in Figure 7

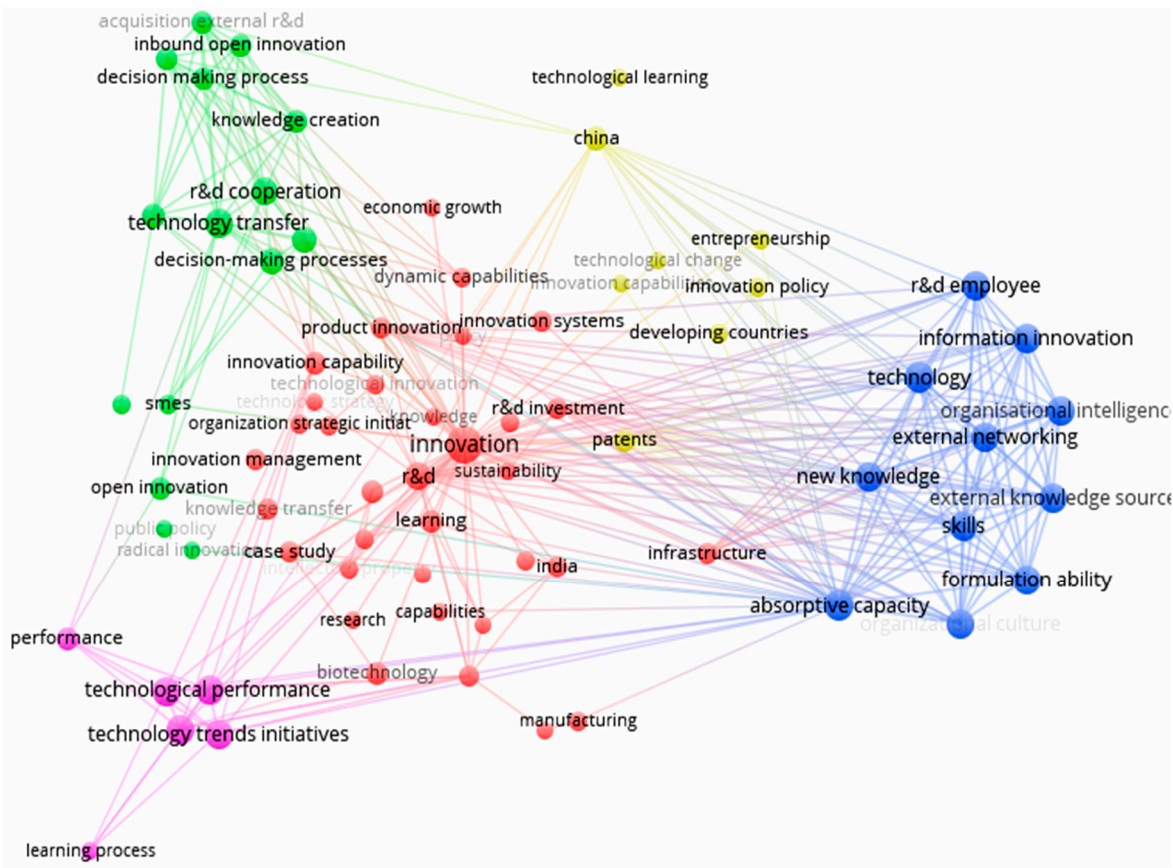

**Figure 7.** VOS viewer Pattern of Innovation Management Capabilities.

It was observed from the huge range of articles selected from 1990 to 2018 that by applying multi-dimensional scaling (MDS) along with burst detection algorithm exploration, around 1627 keywords were screened out. Among the 1627 keywords, 75 words met the minimum threshold of 5 occurrences. Among these 75 words there were a number of emerging and fading themes that appeared that were relevant to the selected research topic [172,173]. In the case of cluster 2; (Green in color) some trends reflected the process aspect of IM capabilities that contributed to their influence on R&D. Some of the keyword trends include R&D cooperation; Acquisition Internal R&D; Acquisition External R&D; Technology Transfer Decision Making process; Knowledge Sharing; Inbound Open Innovation; Project management (control & monitoring) Innovativeness compatibility; Internal & external Knowledge sharing ability; and Open Innovation; Knowledge creation process). Clusters 1 and 5 (Red and pink in color) reflected some of the themes that represent the strategic prospective of IM capabilities. These themes include IP performance, Technological Performance, Innovative Performance, Technology trends, Organization strategy, and Innovation strategies and initiatives. In the case of cluster 3 (Blue in color) infrastructural aspects of IM capabilities were represented. These themes include: R&D investment; External Networking; R&D Employee; New Knowledge; Radical Innovation; External knowledge; Formulation; Absorptive capacity; and Knowledge incentives. The cluster, 4, represents some of the key trends related to effective policy development for technological learning in developing countries.

- Cluster 2 (Green in color)

R&D cooperation; Acquisition Internal R&D; Acquisition External R&D; Technology Transfer; Decision Making process; Knowledge Sharing; Inbound Open Innovation; Project management (control & monitoring) Innovativeness compatibility; Internal & external Knowledge sharing ability; Open Innovation; Knowledge creation process. It is quite interesting that after extensive analysis Cluster 2 seems to be reflects operational aspect of innovation management. Due to the characteristics of occurrence, keywords reflecting innovative capabilities were basically influential on the developing processes for managing innovation

- Cluster 3 (Blue in color)

R&D investment; External Networking; R&D Employee; New Knowledge; Radical Innovation; External knowledge; Formulation; Absorptive capacity; Knowledge incentives. Its seems to be more extensive, as a majority of research studies reflect the infrastructural prospective of innovation management capabilities

- Clusters 1 and 5 (Red and Pink in color)

IP performance, Technological Performance, Innovative Performance, Technology trends, Organization strategy, Innovation strategies and initiatives Similar to the case of Innovation Management, this co-occurrence keyword helps to identify appropriate studies with core concepts of Innovation management capabilities with enablers of the dominance of Innovation capability with influence on R&D. For Innovation Management, co-words and co-occurrence keywords help identify appropriate studies with core concepts of Innovation management capabilities that have influential enablers of the dominance of Innovation management capability with an impact on R&D, as shown in Table 10.

**Table 10.** Innovation Management (I.M) Capabilities dimension.

| Authors | Process | IM Infrastructure | Strategy |
|---|---|---|---|
| Rodriguez and Frank [174] | R&D cooperation; Internal R&D; External R&D | N/A | N/A |
| Chanwoo et al. [175] | N/A | R&D investment; External networking; R&D employee | IP performance; Technological Performance |
| Kondratiuk-Nierodzińska [176] | Technology transfer | New knowledge; Absorptive capacity | N/A |
| GarcÃa-Granero et al. [177] | Decision making process; Internal R&D; External R&D | External knowledge; Formulation | Innovative performance |
| Sáenz et al. [178] | Knowledge sharing (IT, Personal interaction, Embedded in management process) | N/A | N/A |
| Spithoven et al. [179] | Inbound and outbound open innovation | Absorptive capacity | N/A |
| Jain, Karuna, Qutbuddin Siddiquee [58] | Project management (control & monitoring); Innovativeness compatibility; Rate of introduction of new product/service per year; Internal & external Knowledge sharing ability | N/A | Innovation strategies & initiatives; Technology trends assessment |
| Numprasertchai, Somchai, Phasit Kanchanasanpetch [180] | Knowledge creation process | Knowledge incentives | Organization strategy |
| Stüer, Christian, Stefan Hüsig [181] | Inbound open innovation | Radical innovation | N/A |

The selected articles from 1990 to 2018 were illustrated and divided into three different areas of innovation management capabilities (IMC) with respect to their similar characteristics. The first research area is directly referred to as process capabilities, which include Technology transfer, Project management, Decision Making process, Open innovation, knowledge creation process, compatibility and Rate of Introduction of new product. The second research area refers to drive infrastructure

capabilities related to innovation management that are required for the R&D function to strengthen their competencies that interface with existing capabilities. In this manner, publication topics have mainly focused on several different dimensions that directly relate to R&D, such as R&D intensity, External Networking, Employee learning, new knowledge, Absorptive capacity, Formulation, Internal and external knowledge sharing, organization strategy, Incentives, and Knowledge management. The third research areas referred to driving strategic capabilities related to Innovation management, this research area includes Performance, Innovation capability, own R&D function, Innovation strategies Initiative, Technology Assessment, and R&D capabilities. After careful consideration there are some studies that emerge that closely relate to the topics shown in Table 11.

**Table 11.** Internal determinates of I.M dimension.

| | | Enablers | References |
|---|---|---|---|
| **Innovation Management Capability** | **Process** | Technology Transfer | [182–188] |
| | | Decision Making Process | [69,124–191] |
| | | Open Innovation | [69,129–199] |
| | | Project Management | [200–202] |
| | | Innovativeness Compatibility | [58,203–207] |
| | | Knowledge creation process | [208–212] |
| | | R&D Corporation | [175,213–215] |
| | | External R&D Acquisition | [216–219] |
| | | Internal R&D Acquisition | [220–223] |
| | | Knowledge Sharing | [224–230] |
| | | Rate of introduction of new product/ service per year | [58–231] |
| | | Internal & external Knowledge sharing ability | [96–126] |
| **Innovation Management Capability** | **Infrastructure** | External Networking | [232–249] |
| | | R&D Investment | [175,246–248] |
| | | R&D Employee | [175,250–257] |
| | | Absorptive capacity | [258–263] |
| | | New knowledge | [59,176,264–267] |
| | | External Knowledge Acquisition | [232–249] |
| | | Formulation | [177,268–271] |
| | | Knowledge Incentive | [180,272–274] |
| | | Radical Innovation | [275–277] |
| **Innovation Management Capability** | **Strategies** | External Knowledge Source | [126–133] |
| | | Internal Knowledge Source | [126,134–137] |
| | | Joint internal Collaboration | [51–138] |
| | | Joint External Collaboration | [51–146] |
| | | Innovative Performance | [278–282] |
| | | External R&D Function | [216–219] |
| | | Innovation strategies & initiatives | [69,120–261] |
| | | Technology Trends | [58,262–297] |
| | | Organization strategy | [180,298–300] |
| | | Intellectual Property Performance | [301–305] |
| | | Technological performance | [158,170–309] |

*4.5. Assessing Technology Management Capabilities*

In most recent studies, extensive bibliometric analyses related to Technology management (TM) have been performed to represent general trends of TM [309]. However, these studies have been unable to identify the core capabilities that were involved in contributing to their influence on R&D competitiveness. Many studies highlight specific research areas adjacent to technology management. For instance, Culnan [310] applies a co-citation strategy to identify the fundamentals of IS (information system) and canvases the area of research to create resemblance to an information system rather than to organizational learning. Similarly, Karki [311] investigates the pillars of the sociology of science literature and identifies the unique relationship between information scientists and sociologists, who share creative ideas only when they scholarly interact with each other. The study that most discusses the extensive analysis on Technology management (TM) through the bibliometric review is Pilkington 2014, which illustrates the various trends of technology management over the 2007 to 2014 period. Somewhat unpredictably, all this existing literature identifies the utilization of TM with a diverse approach to draw a general perspective of TM; however, they rarely classify the resources that drive capabilities related to Technology Management (TM).

We accepted 13,567 relevant articles by systemically searching the Scopus Database (n = 13,567). After removing research articles that did not fulfill the eligibility criteria based on PRISMA, already discussed in chapter 3, a total of 662 non-duplicate articles were recognized including a number of research articles with author supplied keywords (n = 394) from the period 1990–2018. The number of research studies without Keywords (n = 268) from the period 1990–2018 were also analyzed. Research article analyses on the basis of Tile, Keywords and Abstract. Because of the characteristic of acknowledging the eligibility criteria, we exclude 12,808 articles overall. The comprehensive representation of the record exclusion at each stage is shown in the PRISMA diagram in Figure 8. Similarly, to trace the potential literature on technology management capabilities, a logical configuration of key phrases with a unique typological pattern was employed in the advanced searching string, which was discussed earlier. Research article analyses were carried out on the basis of Tile, Keywords and Abstract. Because of the characteristic of acknowledging the eligibility criteria, we excluded 6852 articles overall.

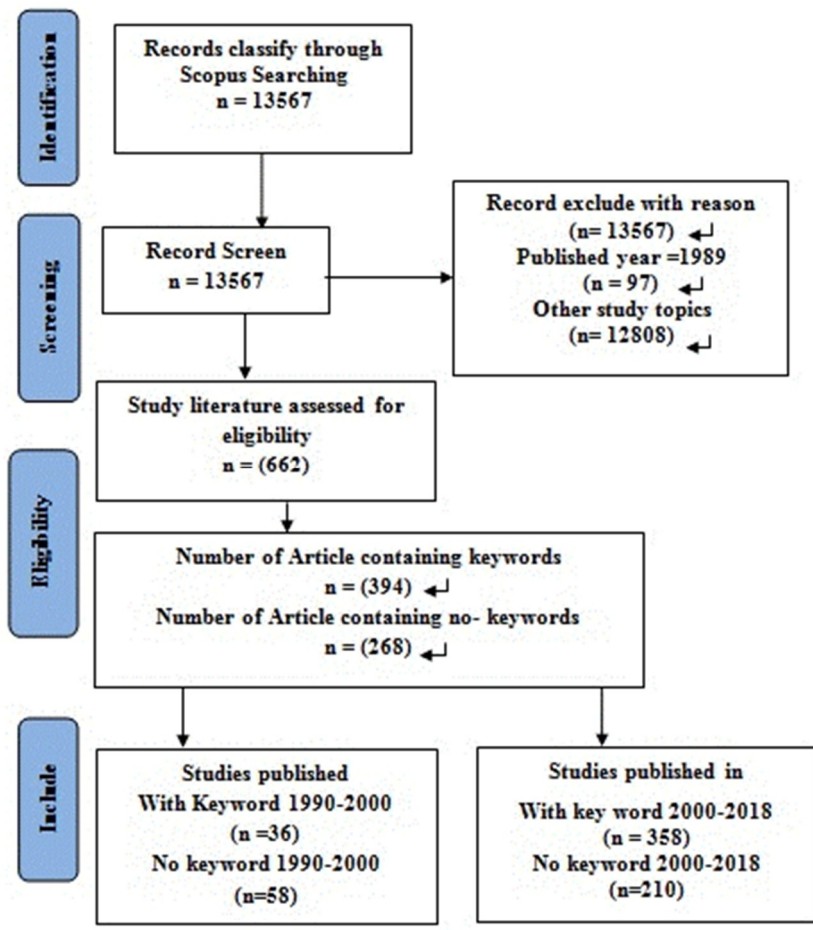

**Figure 8.** PRISMA flow diagram for Technology Management Capabilities during 1990–2018.

The following typology configuration was applied in the case of Technology Management capability on the Scopus search engine: "Technology management" OR "Technology capabilities research capability" OR "Technology capabilities" OR "Technology capacity" OR "Technology Management in R&D" OR "Management and Technology SMEs" OR "TM ability" OR "Technology Management & (R&D) " OR "Technology Strategies OR "Technology capabilities and R&D" OR "Technology Management in research and development" OR "T.M capabilities & (R&D)" OR "TM & R&D capabilities". The complete visual pattern is represented in Figure 9.

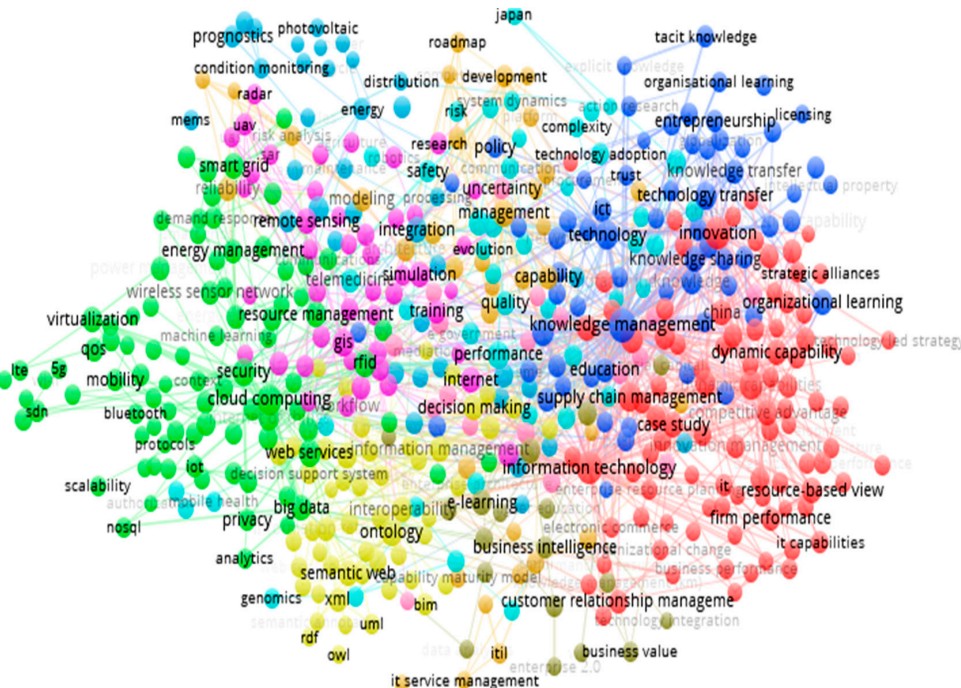

**Figure 9.** The term maps from period of 1990 to 2018.

The 1st cluster allows researchers to explore various study trends related to capabilities that belong to technology management. In this cluster a majority of these studies placed more emphasis on utilizing the capabilities that belong to technology management for developing business strategy and measuring business performance. Effectively utilizing these capabilities allows firms to utilize IT capabilities to enhance their competitive advantage and enables more business diversification. The 2nd cluster includes studies more related to the utilization of technology for privacy and IT securities. A majority of studies illustrate utilizing various IT protocols for secure cloud computing. The 3rd cluster presents some of the overlapping trends regarding the selection of relevant capabilities related to technology management to cater knowledge management for complex R&D. The 4th cluster presents some of the study trends that utilize technological capability for the construction industry. Within the same cluster, there are few studies that highlight the utilization of technological capability for extensive data analysis and data mining. The 5th cluster includes studies in which organizations utilize their technological capabilities for artificial intelligence and data diffusion. Within the same cluster, some studies present extensive utilization of capabilities belonging to technology management for risk and disaster assessment to respond to any crisis situation. The 6th cluster includes some research trends in which a majority of research articles exploit technological capability for telemedicine purposes and for enterprise architecture. In a similar fashion, the 7th cluster shows some research trends that were more aligned towards utilizing technology management capabilities for developing an organizational structure for management systems especially for the health and engineering sector. The 8th cluster includes a range of research studies that present some aspects of technology management capabilities to spread IT governance and services within and beyond organizational boundaries. Within the same cluster there are few studies that represent the role of capabilities related to technology management that allow firms to establish strong software development for the IT industry. The 9th cluster includes a range of research studies that utilize technological capabilities for developing firms' business intelligence. Similarly, the 10th cluster includes a majority of studies that utilize capabilities related to technology management for creating artificial intelligence in the case of developing information security, intelligent databases and for strategic planning. Some of the highest occurrences of key phrases are shown in Table 12.

**Table 12.** Keywords with Highest Occurrence (technology management (TM) capabilities).

| Cluster | Keywords with Highest Occurrence | Cluster | Keywords with Highest Occurrence | Cluster | Keywords with Highest Occurrence |
|---|---|---|---|---|---|
| Cluster 1 | Business strategy, capabilities, Business performance, competitive advantages, customer relationship, diversification, dynamic capabilities, IT capabilities | Cluster 5 | Algorithms, Artificial intelligence, crisis management, data collection, data fusion, disaster management, disaster response, Risk Assessment, Risk management | Cluster 9 | Business intelligence, business value, data analysis, technology integration, information sharing, social media, collaboration, communication management, enterprise management |
| Cluster 2 | Cloud computing, Analytics, computer networks, mobility management, routing and security | Cluster 6 | Electronic health record, Electronic medical records, Enterprise architecture, Evolution, genomics, Telemedicine, Transformation | Cluster 10 | Forecasting, Fuzzy logic, information security, strategic planning, real options, classification, database, intelligent agent |
| Cluster 3 | Adaption, Ambidexterity, concurrent engineering, critical success factor, exploitation, globalization, knowledge Acquisition, knowledge transfer | Cluster 7 | Bio mass, organizational structure, distribution, health management, management systems, measurement; Organizational structure, system engineering, maintenance | | |
| Cluster 4 | Construction management, information management, data analysis, data mining, decision making, distributed system, anthologies | Cluster 8 | Capability, capability maturity mode, computing, development, IT governance, IT services, Knowledge management system, process improvement, reliability, risk analysis, software development | | |

From 35,458 authors, only 2876 key words were supplied that met the threshold with the minimum number of documents equal to 5. The 500 most occurring keywords were divided into 5 clusters; among these clusters, certain keywords directly referred to technology management due to their characteristics. A complete descriptive analysis produced by the VOS viewer software is developed and shown in Table 13.

**Table 13.** Descriptive analysis of technology management capabilities studies concentrated on R&D from 1990–2018.

| Descriptive Analysis 1990–2018 | | |
|---|---|---|
| **Years** | **1990–2000** | **2001–2018** |
| Total Paper | 94 | 568 |
| Minimum no of keywords | 152 | 1129 |
| Minimum occurrence | 3 | 5 |
| Minimum Threshold | 4 | 35 |
| Highest total link strength | 9 | 389 |
| Highest occurrence | 3 | 37 |

*4.6. Emerging and Disappearing Themes (Burst Detection)*

After applying burst detection techniques, 50 emerging and fading themes appeared after exploring, both title and author supplied keywords with respect to time frame are shown in Table 14. The basic mechanism behind the algorithm allows a probabilistic estimation that responds when there is an increasing occurrence of individual words. State switches correspond to the approximate time at which the occurrence of words significantly adjusts.

**Table 14.** Keywords of emergent and fading subjects.

| Latest Bursting and Disappearing Topics | | | | | | | | | | | |
|---|---|---|---|---|---|---|---|---|---|---|---|
| **In the Title** | | | | | | **In the Keywords** | | | | | |
| **Word** | **Level** | **Weight** | **Length** | **Start** | **End** | **Word** | **Level** | **Weight** | **Length** | **Start** | **End** |
| america | 1 | 7.832402 | 3 | 2011 | 2013 | control | 1 | 4.186465 | 1 | 2014 | 2014 |
| volum | 1 | 4.663473 | 2 | 2012 | 2013 | framework | 1 | 2.394371 | 2 | 2015 | |
| util | 1 | 2.377221 | 1 | 2013 | 2013 | design | 1 | 2.111547 | 3 | 2014 | |
| softwar | 1 | 2.205353 | 1 | 2014 | 2014 | led | 1 | 2.577693 | 1 | 2011 | 2011 |
| system | 1 | 4.108759 | 1 | 2013 | 2013 | Technolo breadth | 1 | 2.396732 | 2 | 2013 | 2014 |
| design | 1 | 3.430824 | 2 | 2015 | | emerg | 1 | 2.396732 | 2 | 2013 | 2014 |
| 19th | 1 | 2.535035 | 1 | 2013 | 2013 | govern | 1 | 2.396569 | 1 | 2014 | 2014 |
| 17th | 1 | 2.53569 | 1 | 2011 | 2011 | framework | 1 | 2.101704 | 2 | 2015 | |
| 2011 | 1 | 5.13143 | 1 | 2011 | 2011 | led | 1 | 2.621172 | 1 | 2011 | 2011 |
| compani | 1 | 3.287214 | 1 | 2011 | 2011 | plan | 1 | 2.680235 | 1 | 2013 | 2013 |
| 2010 | 1 | 2.502768 | 1 | 2010 | 2010 | Technoloposture | 1 | 2.396569 | 1 | 2014 | 2014 |
| 2013 | 1 | 6.568964 | 1 | 2013 | 2013 | process | 1 | 4.812644 | 5 | 1998 | 2002 |
| america | 1 | 2.784436 | 3 | 2011 | 2013 | manufacture | 1 | 4.914048 | 4 | 2004 | 2007 |
| decis | 1 | 2.08223 | 1 | 2010 | 2010 | Technol level | 1 | 4.885668 | 2 | 2014 | 2015 |
| amci | 1 | 2.784436 | 3 | 2011 | 2013 | strategi | 1 | 8.30742 | 12 | 1990 | 2001 |
| confer | 1 | 2.12198 | 2 | 2012 | 2013 | electron | 1 | 4.308646 | 12 | 1992 | 2003 |
| inform | 1 | 6.198066 | 1 | 2013 | 2013 | dynam | 1 | 4.732141 | 2 | 2014 | 2015 |
| adopt | 1 | 2.36177 | 2 | 2013 | 2014 | analysi | 1 | 3.732285 | 1 | 2008 | 2008 |
| ici | 1 | 2.21862 | 1 | 2013 | 2013 | area | 1 | 3.043425 | 4 | 2007 | 2010 |
| organis | 1 | 2.273573 | 2 | 2012 | 2013 | busi | 1 | 2.933504 | 3 | 2001 | 2003 |
| role | 1 | 4.4471 | 7 | 2011 | | micro grid | 1 | 3.685219 | 4 | 2014 | |
| busi | 1 | 2.933504 | 3 | 2001 | 2003 | smart | 1 | 2.96581 | 2 | 2016 | |
| capable | 1 | 5.761972 | 3 | 2015 | | Techno timing | 1 | 4.281443 | 1 | 2014 | 2014 |
| product | 1 | 4.999968 | 11 | 1990 | 2000 | virtual | 1 | 3.256993 | 2 | 2011 | 2012 |
| enterprise | 1 | 5.227033 | 5 | 2013 | | execute | 1 | 2.944126 | 3 | 2011 | 2013 |
| design | 1 | 4.710003 | 3 | 2013 | 2015 | leadership | 1 | 2.97387 | 2 | 2012 | 2013 |
| 2012 | 1 | 6.499614 | 1 | 2012 | 2012 | acquisition | 1 | 2.924045 | 3 | 2003 | 2005 |
| 2011 | 1 | 6.32575 | 1 | 2011 | 2011 | storage | 1 | 2.948039 | 4 | 2014 | |
| manufacture | 1 | 8.434381 | 15 | 1985 | 1999 | cloud | 1 | 3.628274 | 5 | 2013 | |
| 2013 | 1 | 7.731247 | 1 | 2013 | 2013 | big | 1 | 2.985318 | 4 | 2014 | |

**Table 14.** *Cont.*

| Latest Bursting and Disappearing Topics | | | | | | | | | | | |
|---|---|---|---|---|---|---|---|---|---|---|---|
| In the Title | | | | | | In the Keywords | | | | | |
| Word | Level | Weight | Length | Start | End | Word | Level | Weight | Length | Start | End |
| america | 1 | 7.832402 | 3 | 2011 | 2013 | control | 1 | 4.186465 | 1 | 2014 | 2014 |
| manag | 1 | 4.454555 | 3 | 1987 | 1989 | brand | 1 | 3.146538 | 2 | 2012 | 2013 |
| inform | 1 | 8.727171 | 1 | 2013 | 2013 | power | 1 | 3.166639 | 4 | 2014 | |
| acquisition | 1 | 4.499209 | 11 | 1996 | 2006 | inform | 1 | 2.842111 | 1 | 2013 | 2013 |
| ici | 1 | 4.052319 | 3 | 2011 | 2013 | adopt | 1 | 2.397288 | 2 | 2013 | 2014 |
| 18th | 1 | 3.887223 | 1 | 2012 | 2012 | organis | 1 | 2.559503 | 2 | 2012 | 2013 |
| volum | 1 | 9.277554 | 4 | 2010 | 2013 | intellig | 1 | 3.170296 | 2 | 2008 | 2009 |
| system | 1 | 7.586908 | 2 | 2012 | 2013 | compani | 1 | 4.643708 | 1 | 2011 | 2011 |
| strateg | 1 | 3.903115 | 1 | 1995 | 1995 | step | 1 | 2.931369 | 4 | 2003 | 2006 |
| innov | 1 | 4.33185 | 2 | 2010 | 2011 | web | 1 | 3.337052 | 6 | 2002 | 2007 |
| electron | 1 | 4.8662 | 13 | 1991 | 2003 | orient | 1 | 2.912103 | 2 | 2006 | 2007 |
| decis | 1 | 5.130489 | 2 | 2010 | 2011 | data | 1 | 3.684005 | 3 | 2012 | 2014 |
| amci | 1 | 7.997926 | 3 | 2011 | 2013 | chang | 1 | 3.339958 | 2 | 2002 | 2003 |
| confer | 1 | 11.78668 | 3 | 2011 | 2013 | program | 1 | 3.620025 | 4 | 2001 | 2004 |
| emerg | 1 | 5.903353 | 4 | 2011 | 2014 | engin | 1 | 4.391811 | 2 | 2007 | 2008 |
| softwar | 1 | 2.183268 | 1 | 2014 | 2014 | captur | 1 | 3.005358 | 1 | 2015 | 2015 |
| compani | 1 | 3.796763 | 1 | 2011 | 2011 | model | 1 | 3.050095 | 1 | 2015 | 2015 |
| decis | 1 | 2.344426 | 1 | 2010 | 2010 | control | 1 | 2.992354 | 1 | 2014 | 2014 |
| | | | | | | mission | 1 | 3.686478 | 3 | 2004 | 2006 |
| | | | | | | outsourc | 1 | 3.964885 | 5 | 2005 | 2009 |
| | | | | | | assess | 1 | 3.492136 | 4 | 2001 | 2004 |

There are certain keywords that were chosen to illustrate the active themes which continue to be included in certain literature related to R&D; such themes include the keywords: Framework (2015-active), design (2014-active), Micro grid (2014-active), Smart (2016-active), storage (2014-Active), cloud (2013-active), Big (2014-active), Power (2014-active), and capability (2015-active). These keywords reflect the probable research themes that are very active in the current research pattern. Conversely, there are certain fading themes that are not significant and that are followed less in contemporary research trends. Such keywords are not included in either the author's supplied keyword list or in the following research titles: dynamic (2014–2015), Analysis (2001–2002), strategies (1990–2001), technology Breadth (2013–2014), emergence (2013-2014), technology level (2014–2015), execution (2011–2013), technology posture (2014–2014), leadership (2012–2013), acquisition (2003–2005), technology timing (2014–2014), model (2015–2015), control(2014–2014), mission (2004–2006), Assess(2001–2004), and outsource (2005–2009).

For technology management, three clusters have been selected from sixteen different clusters due to the frequency of their appearance as shown in Figure 10. The green cluster includes (Technology Acquisition; Technology Exploitation; Technology Learning; Technology Planning; Technology Development; Technology Deployment; Technology Assessment; Technology Forecasting; Technology Watch; Technology research; and Technology Improvement). However, the Red cluster includes (organization capability, Facility; Management capability; personal skill; Structure; and Culture), similarly Blue cluster includes (Internal technology Development; External Technology Collaboration; Normative strategic Technology Management, Strategic Technology Management; and Operative Technology Management; Absorptive capacity; Desorptive capacity).

From the above, the cluster logical traces trend toward the strategic perspective of Technology management in addition to their internal and external dimensions. Occurrences of keywords help to trace the relevant studies through which Technology management capabilities with influential enablers may recognize the potential driving factors behind the dominance of Technology management capability's influential impact on R&D, as shown in Table 15.

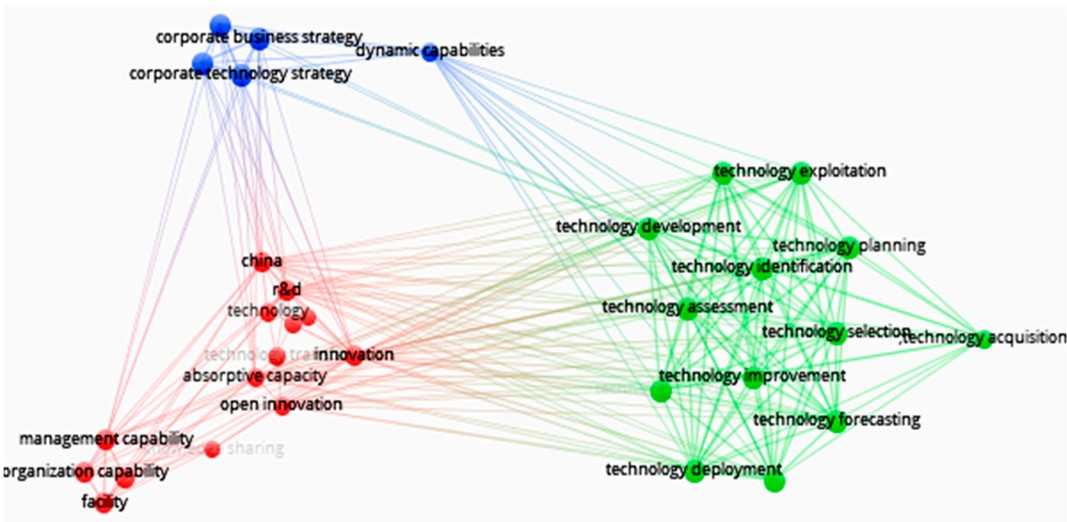

**Figure 10.** VOS viewer Pattern on Technology Management Capabilities.

**Table 15.** TM capabilities dimension.

| Authors | Process | IM Infrastructure | Strategy |
|---|---|---|---|
| Dilek et al. [312] | Technology Acquisition; Technology Exploitation; Technology Identification; Technology Learning; Technology Protection; Technology Selection | N/A | N/A |
| Günther [313] | Technology Planning; Technology Development; Technology Deployment; Technology Protection; Technology Assessment; Technology Forecasting | N/A | N/A |
| Lee [314] | N/A | N/A | Strategic Technology; Road Mapping |
| Zabala and Iturriagagoitia [315] | Technology Watch; Technology Exploiting; Technology Development | N/A | N/A |
| Lee [316] | N/A | N/A | Strategic Technology; Road Mapping |
| Won and Park [317] | N/A | N/A | Strategic Technology; Road Mapping |
| Arasti et al. [318] | N/A | N/A | Corporate Business; Strategic Capability; Corporate Technology; Strategic Capability; Technology Alliance |
| Lichtenthaler et al. [319] | Technology Transfer | N/A | Absorptive Capacity; Descriptive Capacity |
| Cetindamara et al. [312] | Identification; Selection; Acquisition; Exploitation; Protection; Learning | Management Competency; Facility; Organization Potential; Personal Skill | N/A |
| Jun and Maximilian [320] | Technology Acquisition; Technology Assimilation; Technology Improvement | N/A | N/A |

Similarly, for technology management after evaluation, ten studies were selected that revealed three sets of TM capabilities. These three capability sets are classified as follows: (1) process, (2) infrastructure, and (3) strategic capabilities. The Technology management process capability consists of numerous processes within functional units of an organization. The basic scope of technology management is far wider than the aspects that directly interface during process innovation and R&D. Similarly, (2) Infrastructure capabilities are recognized as an essential contributor to the knowledge-oriented economy. To construct and utilize new knowledge, the sharing of information within the existing knowledge needs to be supported by integrating the different technological platforms. However, (3) strategic capabilities should not be created alone independent of the existing business strategy; relative technological assets should be recognized as major components of business planning. Therefore, the comprehensive driving factors are represented in Table 16.

**Table 16.** Internal determinates of T.M dimension.

| | | Enablers | References |
|---|---|---|---|
| **Technology Management Capability** | **Process** | Technology Acquisition | [312,320–323] |
| | | Technology Exploitation | [44,324,325] |
| | | Technology Identification | [44–330] |
| | | Technology Learning | [312,331,332] |
| | | Technology Planning | [333–336] |
| | | Technology Development | [44–340] |
| | | Technology Selection | [341–344] |
| | | Technology Deployment | [313,345,346] |
| | | Technology Assessment | [328,347–349] |
| | | Technology Forecasting | [350–355] |
| | | Technology Watch | [76,315,356–359] |
| | | Technology Assimilation | [360–363] |
| | | Technology Research | [328,364] |
| | | Technology Protection | [26,44,365] |
| | | Technology Improvement | [320,366–368] |
| **Innovation Management Capability** | **Infrastructure** | Management Competency | [26,369–371] |
| | | Organizational Potential | [372–375] |
| | | Facility and Equipment Capability | [26,376,377] |
| | | Personal Skill | [26,378–384] |
| **Innovation Management Capability** | **Strategies** | Desorptive Capacity | [257–385] |
| | | T.M (Corporate Technology Strategy) | [10,44,288–390] |
| | | T.M(Corporate Business Strategy) | [10,318,391] |
| | | T.M(Technology Alliance Strategy) | [387,392,393] |
| | | Strategic Technology Road Mapping | [312,314–396] |
| | | Absorptive Capacity | [385,397–399] |

## 5. Conceptual Framework

The conceptual theory is represented in this research based on theoretical evidence that highlights the significant description with a relevant clarification of vital conditions that influence the research breakthrough. These concepts influence certain factors that are crucial in supporting a hypothetical argument. Furthermore, the conceptual Model consists of certain dimensions that justify the crucial conditions and are conceived as imperative for estimating a logical interpretation for developing practical relevancy. The conceptual model suggested in this research assesses the potential capabilities that directly influence the generic R&D characteristics, that is, the conceptual framework that is not limited to specific R&D. Therefore, the framework can be applied to assess any R&D operations. The author is aware of the reality that the significant output during the assessment depends upon three influential factors: Technology, knowledge and Innovation management and unusual vital interrelation conditions, which were also observed during the systematic review of the literature.

The conceptualization of KM capabilities overlaps to an extent with IM capabilities, particularly in addressing the R&D context. The innovation approach always encourages the resolution of complex problems, adds values and develops organizational competencies to be generated as an outcome of a comprehensive practice of knowledge capability [319]. Certain basic concepts of innovational capabilities heavily interrelate with knowledge capabilities and overlap specifically when addressing the concept of intellectual property as a vital source of innovation and of acquiring new knowledge for rapid growth [400]. In general, understanding organizational knowledge capabilities means interpreting the potential organizational capability of the firms, which encourages the implementation of the mechanism to respond regarding what must be done before their business rivals do it, by developing and managing the existing innovation capability according to the requirements [401].

In recent decades, a sequence of questions and confrontation with potential criticisms have arisen regarding the value creation and an effectiveness of selecting relevant technology for catering knowledge management capabilities [402]. To address the dynamic business environment, existing knowledge capability needs to be redesigned in the context of cross-functional communication and external collaboration with a changing approach to existing technological capabilities [403,404]. Accorsi (2008) describes one of the most extensively available technological capabilities as an instrument to develop knowledge capability is known as a Knowledge Management System (KMS). Venters (2010)

suggested a variety of Technological capabilities that have been involving for precisely in reshape the potential attribute of knowledge management capabilities [405].

Innovation management and technology management are now considered research areas under the rubric of management. The most decisive target for achieving the business competency for any research based organization in terms of extending their footprint on market is completely depends on interconnectivity among capabilities related to technology management and innovation management. According to the business environment research based organizations consistently updating their innovative and technical ability which comprises consistent estimation, continuous monitoring, and developing techno innovational capabilities [406].

Prior studies mostly highlighted the relationships between knowledge, innovation and technology management at an individual level and drew a capability perspective that influences R&D. However, these studies are somehow unclear regarding drawing a relationship between the three concepts together. Therefore, to address this gap the author performs a comprehensive systematic review in order to propose a conceptual framework that illustrates the relationship between the three sets of capabilities all together as shown in Figure 11.

*Limitations and Future Research Aveneue*

Certain limited and suitable deviations from normal research opportunities in uncertain situations are highly accepted. However, this deviation is not a justification for poor oversight of other influential factors that affect R&D activities. New guidelines are required to better determine other dimensions that involve three sets of capabilities. Further systematic review should be conducted to extract any novel criteria that are needed to assess whether similar findings arise. In the case of future avenues, to keep the model clear, we only include the capabilities with their relevant resources. These capabilities draw process, infrastructure and strategic aspects that have been shown to be important in predicting resources for effective R&D outcomes. The conceptual model may allow researchers to develop complex approaches such as augmented reality (AR), SLAM (simultaneous localization and Mapping) and deep learning to design advanced R&D for sensitive technologies like diagnosing brain tumors using big data from the medical Internet of Things, object-centric data management and visualization for augmented reality, or developing deep learning algorithms for predicting dosages for the treatment of modalities in radiation therapy for cancer patients.

## 6. Conclusions

The sudden expansion of global business competition with complex R&D challenges compresses firms and stifles their capacity for creativeness and innovativeness. This extension requires accurate judgment to spend more on the interdependent elements among knowledge, innovation and technology management capability. The recommendations from this review are threefold. First, a conceptual vision was produced by connecting insights from the in-depth analysis of a systematic review with a theoretical foundation that directs empirical research. This comprehensive research review helps policy makers analyze interlinks between the criteria and sub-criteria of each dimension that are traced from the previous literature and sets the boundaries between knowledge, innovation and technology management capabilities. Second, from a novel perspective the process, infrastructure and strategic parameters of each of the three sets of supporting management capabilities by improving and developing the strength of organizational knowledge, Innovation and Technology management capabilities based on the criteria can be achieved. These capabilities were comprehensively proven by prior studies to depend on the process, the infrastructure and the strategy. Lastly, this research article proposed a theoretical framework through the systemic analysis of the potential directions of R&D management. This article encourages placing the diverse streams of capabilities into different operational categories and provides a comprehensive overview of research and development regarding the extent to which all three sets of capabilities can influence R&D activities to avoid market dynamism.

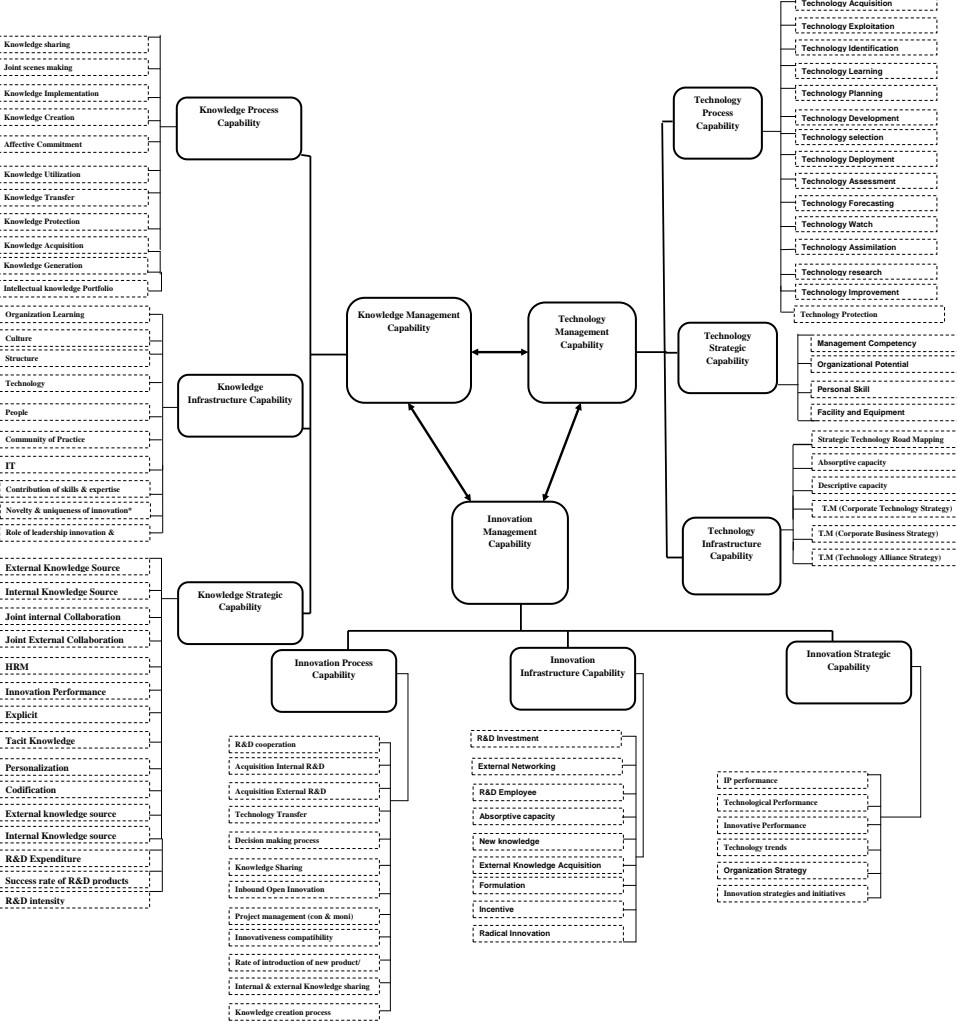

**Figure 11.** Conceptual framework.

**Author Contributions:** Z.A. is a PhD Scholar in Universiti Malaysia Pahang. His research is in the field of R&D and Engineering management. He has a few published articles in his field. S.S. is an associate professor in Universiti Malaysia Pahang. His research area is management sciences. He has a few published books and articles in his field.

**Funding:** This research received no external funding.

**Acknowledgments:** This works administratively and technically supported by *Asian Science Consortium* for their useful suggestion

**Conflicts of Interest:** The authors declare no conflict of interest.

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
