# Peer review of "Exploring the Role of Knowledge, Innovation and Technology Management (KNIT) Capabilities that Influence Research and Development"

_2199-8531, doi:10.3390/joitmc5020021_

Round 1
Reviewer 1 Report
A very important topic, closely related to the profile of the magazine. .Very good review article. I do not make any comments.
Author Response
Thanks for evaluating my work. It help me to improve my work under your thoughtful comments please accept my correction
kind regard
zeeshan Asim

Reviewer 2 Report
The topic of KNIT is possible to consider as up-to-date, because of the influence of globalization on all companies. In the text, I found several errors, which need to be correct.
Chapter 2. Method is unclearly described. It is difficult to find methods which are applied on the way of getting relevant data.
Chapters 2.3 and 2.4 should be rewritten from "word-bullets" to plain text. Some 2.x chapters are not necessary to be in separate with only one sentence (e.g. 2.5, 2.7.3., 2.7.5 or 2.7.7). There should be better to merge into a bigger part (according to the meaning of the merged text).
There is a vague description of keywords, which were used for searching in the Scopus database. If authors used specific keywords (such in table 2, 8, 14), they could refer to these tables in the text. Therefore, there are difficult to understand data in figures 2, 5, 8. All of these three figures include numbers of articles founded in Scopus, but no description, how they found them.
Between chapter title and table/picture must be descriptive text, connected to the object.
In tables 4, 6, 10, 12 there are doubled references in some rows. The form of quotations in these tables could be rewritten into one style. I do not understand, why tables 10-12 are divided into three separate parts.
Authors made some formal errors in the numbering of chapters. Therefore, there are two Discussion chapters (page 19 and page28).
Author Response
Thanks for evaluating my work. Its help me to improve my work under your thoughtful comments please accept my correction
kind regard
Zeeshan Asim

Reviewer 3 Report
Overall, this is an interesting piece of research. I have two major concerns and one minor:
Minor: I think Headline 1.1 is too much slang.
Major 1: Readability; please use a professional copy editor.
Major 2: The methods / review is very well conducted. However, after reading it, I qasked myself "so what". What did I learn and how does it impact understanding and the future of knowledge management? Maybe you can be a bit more "futuristic" or "philosophical" in the backend of the paper. consider, for example, the impact of big data or augmented reality. I did a quick look on google search and found some interesting chapters on knowledge management and AR / Enterprise social networks (e.g. https://link.springer.com/chapter/10.1007/978-3-658-12652-0_5 ). This could make your paper more interesting and create more value to readers. It could also inspire future research. I suggest you extend your discussion.
Thanks for evaluating my work. It help me to improve my work under your thoughtful comments please accept my correction
kind regard
Zeeshan Asim

Round 2
Reviewer 2 Report
The topic of knowledge, innovation and technology management (KNIT) could be described as one of the most important field, on which companies must to focus on up-to-date. There is impact of the global entrepreneurs‘ environment because of the globalization.
Author Response
Reviewer 2 Report (second Round) | Author correction |
The topic of knowledge, innovation and technology management (KNIT) could be described as one of the most important field, on which companies must to focus on up-to-date. There is impact of the global entrepreneurs ‘environment because of the globalization
| Edited: Page 2 Paragraph no 2 Line number 10-16 Paragraph no 3 line number 17-31 |
